# Electronic-reconstruction-enhanced hydrogen evolution catalysis in oxide polymorphs

Yangyang Li[1], Zhi Gen Yu[2], Ling Wang[1], Yakui Weng[3], Chi Sin Tang[4,5], Xinmao Yin[4,6], Kun Han[7], Haijun Wu[1], Xiaojiang Yu[6], Lai Mun Wong[8], Dongyang Wan[9], Xiao Renshaw Wang[7], Jianwei Chai[8], Yong-Wei Zhang[2], Shijie Wang[8], John Wang[1], Andrew T.S. Wee[4,5], Mark B.H. Breese[6], Stephen J. Pennycook[1,5,9], Thirumalai Venkatesan[1,4,5,9], Shuai Dong[10], Jun Min Xue[1] & Jingsheng Chen[1]

Transition metal oxides exhibit strong structure-property correlations, which has been extensively investigated and utilized for achieving efficient oxygen electrocatalysts. However, high-performance oxide-based electrocatalysts for hydrogen evolution are quite limited, and the mechanism still remains elusive. Here we demonstrate the strong correlations between the electronic structure and hydrogen electrocatalytic activity within a single oxide system $Ti_2O_3$. Taking advantage of the epitaxial stabilization, the polymorphism of $Ti_2O_3$ is extended by stabilizing bulk-absent polymorphs in the film-form. Electronic reconstructions are realized in the bulk-absent $Ti_2O_3$ polymorphs, which are further correlated to their electrocatalytic activity. We identify that smaller charge-transfer energy leads to a substantial enhancement in the electrocatalytic efficiency with stronger hybridization of Ti 3$d$ and O 2$p$ orbitals. Our study highlights the importance of the electronic structures on the hydrogen evolution activity of oxide electrocatalysts, and also provides a strategy to achieve efficient oxide-based hydrogen electrocatalysts by epitaxial stabilization of bulk-absent polymorphs.

[1] Department of Materials Science and Engineering, National University of Singapore, Singapore 117575, Singapore. [2] Institute of High Performance Computing, Singapore 138632, Singapore. [3] School of Science, Nanjing University of Posts and Telecommunications (NUPT), Nanjing 210023, China. [4] Department of Physics, Faculty of Science, National University of Singapore, Singapore 117542, Singapore. [5] NUS Graduate School for Integrative Sciences and Engineering, National University of Singapore, Singapore 117456, Singapore. [6] Singapore Synchrotron Light Source, National University of Singapore, 5 Research Link, Singapore 117603, Singapore. [7] School of Physical and Mathematical Sciences & School of Electrical and Electronic Engineering, Nanyang Technological University, Singapore 639798, Singapore. [8] Institute of Materials Research and Engineering, A*STAR (Agency for Science, Technology and Research), #08-03, 2 Fusionopolis Way, Innovis 138634, Singapore. [9] NUSNNI-NanoCore, National University of Singapore, Singapore 117411, Singapore. [10] School of Physics, Southeast University, Nanjing 211189, China. Correspondence and requests for materials should be addressed to J.C. (email: msecj@nus.edu.sg)

With the growing concerns about the environmental pollution, global warming and the rapid depletion of petroleum resources, exploiting alternatively sustainable, clean, and renewable energy sources is becoming the most urgent scientific challenge for us in the modern society[1,2]. Supplying energy without toxic emissions, hydrogen, as a promising energy carrier, is believed to have a crucial role in the future scenario of energy applications[3]. In the past several decades, numerous efforts have been devoted to developing more sustainable hydrogen-production routes from renewable energy sources[3,4]. Among them, electrochemical water splitting using the hydrogen evolution reaction (HER, $2H^+ + 2e^- \rightarrow H_2$) is considered as a promising method for hydrogen production[3–5], owing to its unparalleled capacity and carbon-free nature. Thus, the efficiency of hydrogen production through electrolysis mainly depends on the catalytic performance of the HER electrocatalysts. Currently, the state-of-the-art electrocatalyst for HER is Pt/C, but it suffers from the prohibitive cost and scarcity[2]. Hence, it is highly desirable to explore new efficient HER electrocatalysts with low-cost and earth-abundant elements.

Transition metal oxides (TMOs) have attracted great interest in both condensed matter physics and materials science due to their fascinating tunable physical and chemical properties with high stability, low cost, and environmental friendliness[6–8]. Titanium dioxide ($TiO_2$) is one of the most studied TMOs, and has been extensively explored as a photocatalyst for water splitting. Its efficiency can be dramatically enhanced by increasing its light absorption via fabricating black hydrogenated $TiO_2$ nanocrystals[9,10]. The perovskite oxides with a general formula of $ABO_3$ have been well studied as electrocatalysts for water splitting, and high-performance oxygen evolution reaction (OER) was realized in them[11–14]. Then, substantial efforts were devoted to understanding its mechanism and investigate the fundamental parameters that govern the catalytic activity[15–20]. Consequently, tuning the electronic structures of the electrocatalysts has been considered as an efficient method to enhance the OER activity of TMOs with stronger metal–oxygen (M–O) hybridization or higher covalency of the M–O bonds[12,20,21]. For HER, pure oxides were usually inactive, because of the extremely strong hydrogen adsorption on oxygen atoms. Nevertheless, most recently, several oxides (such as $Pr_{0.5}(Ba_{0.5}Sr_{0.5})_{0.5}Co_{0.8}Fe_{0.2}O_{3-\delta}$[2], $SrNb_{0.1}Co_{0.7-}Fe_{0.2}O_{3-\delta}$ (nanorods)[22], and CoO (nanorods)[23]) were reported to exhibit efficient HER activities. However, the origin of TMO-based HER mechanism with oxides' characteristics, e.g., metal–oxygen hybridization, and the fundamental parameters that dominate the HER activity still remains unclear.

In this work, we demonstrate the close correlation between the electronic structures and HER catalytic activities in a single-oxide system ($Ti_2O_3$) with strongly correlated electrons. Generally, $Ti_2O_3$ has a corundum structure[24,25] in bulk with an ultra-narrow bandgap ($E_g \approx 0.1$ eV), exhibiting excellent photothermal effect[26] and mid-infrared photodetection[27]. Interestingly, a new orthorhombic $Ti_2O_3$ polymorph that is absent in bulk was stabilized on $Al_2O_3$ single-crystal substrates via epitaxial stabilization, with intriguing emergent properties[28,29]. Here, extension of the polymorphism of $Ti_2O_3$ is further achieved by epitaxially stabilizing a cubic phase, which has not been reported before and is also bulk-absent. Taking advantages of the excellent structural flexibility of $Ti_2O_3$, we are able to explore the correlation between the physical electronic structures and HER activities within a single-material system, which provides a more precise understanding on the correlation without disturbance of varied elements. Systematical investigations are performed on the electronic structures, transport properties, and HER activities of three $Ti_2O_3$ polymorphs. Evident electronic reconstructions are observed in the epitaxially stabilized orthorhombic and cubic phases with modulated electron–electron interactions and charge-transfer energy. Importantly, smaller charge-transfer energy leads to the stronger hybridization strength of the Ti 3d –O 2p orbitals, which lower the d-band center of Ti and weakens the H adsorption, further resulting in the enhanced HER activity in those epitaxially stabilized polymorphs. This electronic-reconstruction enhanced HER activity, achieved in the structure-tailored $Ti_2O_3$, introduces a previously unrecognized route to explore more efficient TMO-based HER electrocatalysts by enhancing the metal–oxygen hybridization via selective stabilization of polymorph phases.

## Results

**Fabrication and structural characterizations of $Ti_2O_3$ polymorphs.** Three $Ti_2O_3$ polymorphs were successfully fabricated in the epitaxial film form by using the pulsed laser deposition (PLD) technique (Fig. 1a). During the film growth, the same corundum (trigonal) $Ti_2O_3$ target was used for stabilizing all $Ti_2O_3$ polymorphs. The phase separation was controlled by carefully varying deposition temperature and substrates' symmetry, which are widely used for the oxides epitaxial stabilization[30,31]. With the advent of thin-film epitaxial stabilization, new polymorphs that do not exist in bulk could be stabilized on single-crystal substrates[28,29,31], resulting in the extension of the oxides' polymorphism. Polymorphism describes the occurrence of different lattice structures and symmetries in a crystalline material with identical composition, which is a critical and attractive phenomenon in material science and condensed matter physics[32,33]. Polymorphs would exhibit different physical, chemical, and mechanical properties due to their varied structural characteristics[34–37], e.g., the polymorph-dependent metal–semiconductor transitions in $Ti_3O_5$[38,39].

By recrystallization on the substrates during epitaxial growth (Fig. 1a), three different $Ti_2O_3$ polymorphs (Fig. 1b), trigonal (denoted as α-$Ti_2O_3$), orthorhombic (denoted as o-$Ti_2O_3$), and cubic (denoted as γ-$Ti_2O_3$), are obtained on single-crystalline α-$Al_2O_3$ and $SrTiO_3$ (STO) substrates. More details of the unit cell parameters for $Ti_2O_3$ polymorphs can be found in Supplementary Fig. 1. Both α- and o-$Ti_2O_3$ are stabilized on α-$Al_2O_3$ (0001) substrates, selected by increasing the deposition temperature from 600 to 900 °C, while cubic γ-$Ti_2O_3$ is stabilized on STO substrates at 600 °C. More details for the stabilization of $Ti_2O_3$ polymorphs can be found in the Supplementary Fig. 2 and the Supplementary Note 1. Atomic force microscopy (AFM) was used to characterize the surface microstructure and roughness of the $Ti_2O_3$ films. As shown in Supplementary Fig. 3, the root mean square (RMS) roughness for all $Ti_2O_3$ samples is quite similar and close to 1 nm, which is very small with respect to their thickness (~300 nm), indicating the flat surfaces that are further confirmed by the scanning electron microscopy (SEM) and scanning transmission electron microscopy (STEM) images (Supplementary Fig. 4). Interestingly, unexpected fascinating properties (such as ferromagnetism[28] and superconductivity[29]) were discovered in the o-$Ti_2O_3$, demonstrating strong structure–property correlations in $Ti_2O_3$, which motivated us to further explore the polymorphism of $Ti_2O_3$.

Figure 1c–e shows the in-plane epitaxial relationships for γ-, o-, and α-$Ti_2O_3$ film/substrate heterostructures, respectively. The corresponding high-resolution X-ray diffraction (XRD) patterns are shown in Fig. 1f. The epitaxial relations for α- and o-$Ti_2O_3$ have been determined to be (0001) α-$Ti_2O_3$ || (0001) α-$Al_2O_3$ and (011) o-$Ti_2O_3$ || (0001) α-$Al_2O_3$ by high-resolution STEM and selected area electron diffraction (SAED)[29]. Since γ-$Ti_2O_3$ is a newly stabilized polymorph, we utilized HR-XRD and HR-STEM (Fig. 2) to investigate its epitaxial growth on STO. As shown in Fig. 1f, (004) γ-$Ti_2O_3$ was directly grown on (002) STO

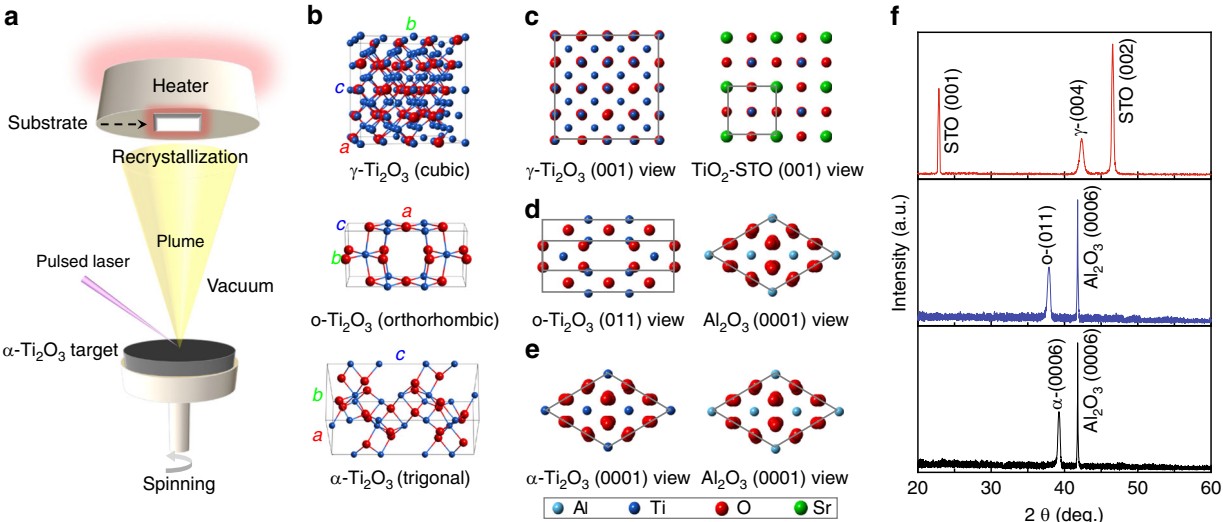

**Fig. 1** Fabrication and structural characterizations of the Ti$_2$O$_3$ polymorphs. **a** Schematic of the PLD chamber where Ti$_2$O$_3$ polymorphs were fabricated using the same (α-Ti$_2$O$_3$) target. **b** Unit cells for γ-Ti$_2$O$_3$, o-Ti$_2$O$_3$, and α-Ti$_2$O$_3$ polymorphs from top to bottom, respectively. **c–e** In-plane epitaxial relationships for γ-Ti$_2$O$_3$ on STO, o-Ti$_2$O$_3$, and α-Ti$_2$O$_3$ on Al$_2$O$_3$, respectively. TiO$_2$–STO (001) view refers to view the TiO$_2$-terminated STO (Supplementary Fig. 5) from the <001> direction. **f** HR-XRD patterns of Ti$_2$O$_3$ polymorphs epitaxially stabilized on the single-crystal substrates. Source data are provided as a Source Data file

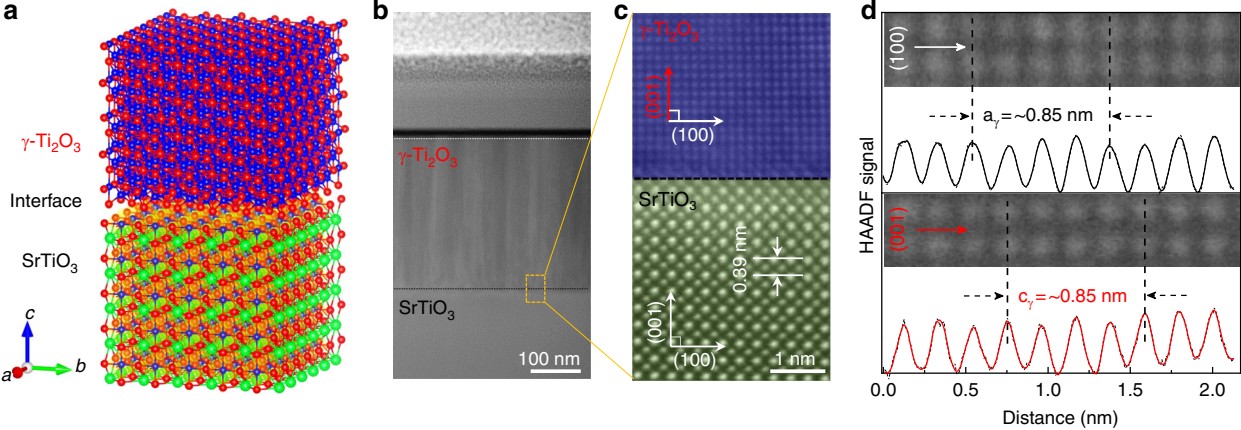

**Fig. 2** Microstructural characterization of the cubic γ-Ti$_2$O$_3$ polymorph. **a** Sketch of the γ-Ti$_2$O$_3$/STO heterostructure. **b** Cross-sectional low-resolution STEM image and (**c**) high-resolution HAADF-STEM image of the γ-Ti$_2$O$_3$/STO interface. **d** HAADF-STEM image and the corresponding HAADF signal profile obtained along (100) and (001) directions of γ-Ti$_2$O$_3$ from (**c**)

(Supplementary Fig. 5), indicating a cubic-on-cubic growth. Figure 2a schematically illustrates the epitaxial growth of γ-Ti$_2$O$_3$ on STO, which is further confirmed by in-plane synchrotron-based X-ray φ-scan (Supplementary Fig. 6 and Supplementary Note 2). As isomorphs of γ-Ti$_2$O$_3$, γ-Al$_2$O$_3$[40,41], and γ-Fe$_2$O$_3$[42] were also stabilized on STO with the same epitaxial relationship. Moreover, γ-Ti$_2$O$_3$ is also successfully stabilized on LaAlO$_3$ (LAO) substrates, which is identical with that on STO (Supplementary Fig. 6).

In-depth STEM measurements were performed to investigate the cubic γ-Ti$_2$O$_3$ on STO. Figure 2b shows the cross-section low-resolution STEM image of the spinel/perovskite γ-Ti$_2$O$_3$/STO heterointerface. The thickness of the γ-Ti$_2$O$_3$ film is about ~300 nm. It needs to be pointed out that the thickness for all Ti$_2$O$_3$ films is fixed at around ~300 nm in order to get the bulk intrinsic properties of these polymorphs. High-resolution high-angle annular dark-field (HAADF) STEM imaging (Fig. 2c) presents the highly crystalline γ-Ti$_2$O$_3$ and the cubic-on-cubic epitaxial

growth on STO. The lattice parameters of γ-Ti$_2$O$_3$ measured from the HAADF-STEM signal profiles (Fig. 2d) along (100) and (001) are a = c ≈ 8.5 Å, consistent with the HR-XRD result ((004): 42.35˚ corresponds to c = 8.53 Å). To shed light on the evolution of Ti valence states at the interface, electron energy loss spectroscopy (EELS) line scan was collected across the γ-Ti$_2$O$_3$/STO interface. As shown in Supplementary Fig. 7, the valence state of Ti evolves from 4 + to 3 + crossing the interface from STO to γ-Ti$_2$O$_3$ with obvious variation at the Ti $L_{2,3}$-edge and O $K$-edge (Supplementary Note 3). Moreover, the Ti $L_{2,3}$-edge EELS spectrum collected from γ-Ti$_2$O$_3$ is same as those obtained from α- and o-Ti$_2$O$_3$[28,43], indicating the same Ti$^{3+}$ chemical environment. Three Ti$_2$O$_3$ polymorphs with same Ti$^{3+}$ valence states but different lattice symmetries are identified and confirmed. Subsequently, the electronic band structures of these Ti$_2$O$_3$ polymorphs are carefully investigated by spectroscopic techniques. In order to eliminate the influence of STO substrates with possible oxygen vacancies, formed during the deposition, that

would affect that light absorption and conductivity of STO[44], the following measurements of γ-$Ti_2O_3$ were mainly performed on the γ-$Ti_2O_3$/LAO samples.

**Determination of electronic reconstructions in $Ti_2O_3$ polymorphs.** With an unpaired $3d^1$ election, α-$Ti_2O_3$ is regarded as an antiferromagnetic Mott insulator with a unique and broad metal–insulator transition (MIT)[45–47]. In strongly correlated TMO systems, the electron–electron interaction, represented by the on-site Coulomb repulsion $U$, is significant, and opens the Mott–Hubbard gap inside the transition metal $3d$ band between the lower Hubbard band (LHB) and upper Hubbard band (UHB). Charge-transfer energy $\Delta$, the energy difference between the oxygen $2p$ band and transition metal $3d$ band, is another crucial parameter in determining the physical and chemical properties of TMOs[20,48]. As illustrated in Fig. 3a, depending on the relative magnitudes of $U$ and $\Delta$, strongly correlated TMOs could be classified into Mott insulators ($U < \Delta$) and charge-transfer insulators ($U > \Delta$). With tunable $U$ and $\Delta$, modulations of the electronic properties that impact on electrochemical activities could be achieved[49–51].

Proposed by Goodenough et al.[52], the electronic band structure of α-$Ti_2O_3$ could be well described using the molecular orbital theory (Fig. 3b). The trigonal crystal field splits Ti $3d$ $t_{2g}$ orbitals into a pair of $e_g^\pi$ orbitals and an $a_{1g}$ orbital near the Fermi level. Furthermore, the $a_{1g}$ band splits into the bonding $a_{1g}$ and the antibonding $a_{1g}^*$ bands, while the $e_g^\pi$ band splits into the bonding $e_g^\pi$ and the antibonding $e_g^{\pi*}$ bands, mainly because of the hybridization of the Ti–Ti orbitals. (The $e_g^\pi$ and $e_g^{\pi*}$ bands are usually located at the same energy level, since the $e_g^\pi - e_g^{\pi*}$

splitting is very small.) The MIT of $Ti_2O_3$ could be explained by the band-crossing scenario[53,54]. That is, the metallic state emerges when the $e_g^\pi$ and $a_{1g}$ bands overlap with increased temperature, while the insulating state occurs when a gap arises between the $e_g^\pi$ and $a_{1g}$ bands with decreased temperature[54]. At room temperature, the bandgap of insulating α-$Ti_2O_3$ is around 0.1 eV, which was verified by both electrical and optical measurements[24,28]. Based on the above proposed electronic structure, the $U$ and $\Delta$ in $Ti_2O_3$ could be assigned specifically to the $a_{1g} - e_g^\pi$ and $O_{2p} - e_g^\pi$ transitions, respectively.

To quantify the $U$ and $\Delta$ in different $Ti_2O_3$ polymorphs, spectroscopic ellipsometry and light absorption measurements were performed at room temperature. Figure 3c–e shows the optical conductivity (σ) of α-, o-, and γ-$Ti_2O_3$, respectively, demonstrating the $a_{1g} - e_g^\pi$ and $a_{1g} - a_{1g}^*$ interband transitions are located at around 1 and 3 eV[55]. Concomitant with the different lattice symmetries, substantial electronic reconstructions were observed among the three $Ti_2O_3$ polymorphs. Specifically, the $U$ ($a_{1g} - e_g^\pi$) in α-, o-, and γ-$Ti_2O_3$ polymorphs increase from 0.85 eV to 1.01 eV, then to 1.10 eV, respectively, revealing stronger electron–electron interactions. Moreover, typical Drude absorptions are observed in o- and γ-$Ti_2O_3$ at low photon energies (ℏω), indicating high free electron concentrations in these two polymorphs. This point will be discussed later, combined with the electronic transport results. The $\Delta$ ($O_{2p} - e_g^\pi$) in the $Ti_2O_3$ polymorphs were obtained from the UV–Vis light absorption data, which are correspondingly shown as insets of Fig. 3c–e. The absorption coefficients (α) of $Ti_2O_3$ polymorphs are in the range of ~0.6 to $2.1 \times 10^5$ cm$^{-1}$ at the UV–Vis range, demonstrating a strong light absorption. In order to extract the

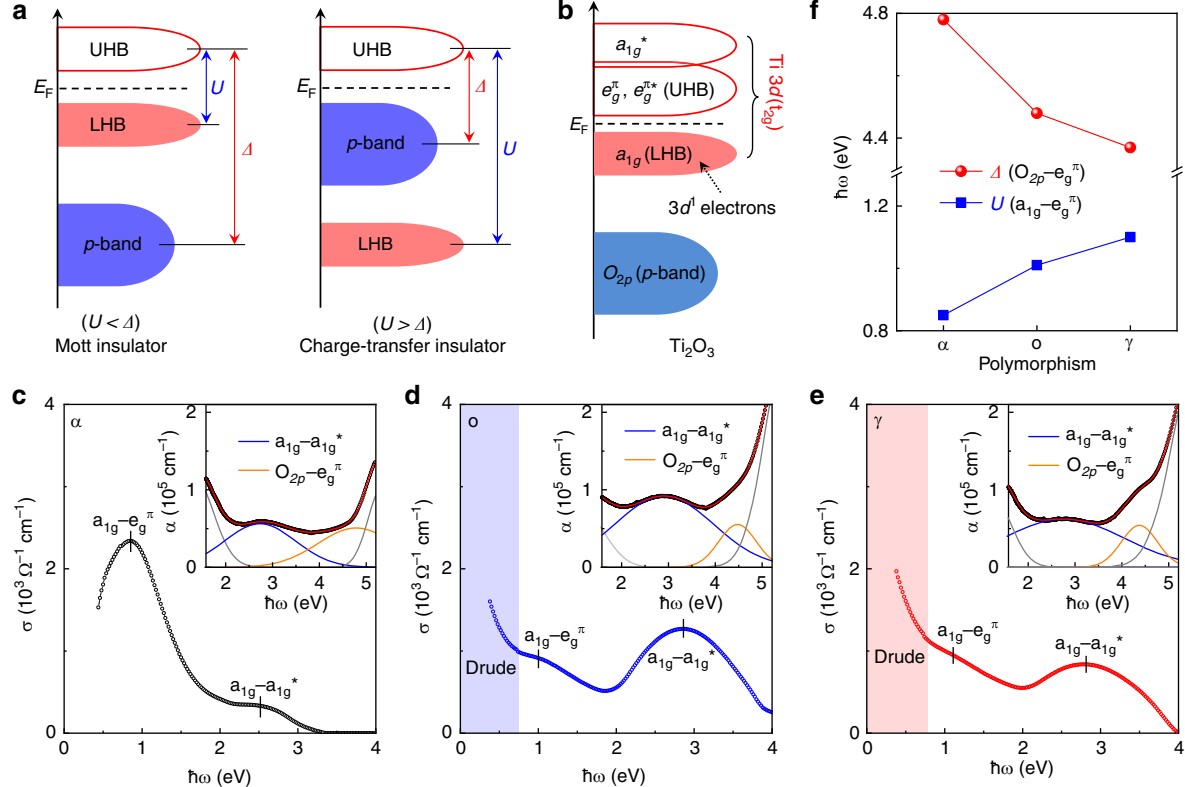

**Fig. 3** Electronic reconstructions in $Ti_2O_3$ polymorphs. **a** Schematic energy band diagram for the Mott insulator and charge-transfer insulator. **b** Proposed electronic structure of $Ti_2O_3$ by Goodenough et al.[52] **c–e** Optical conductivity spectra of the α-, o-, and γ-$Ti_2O_3$ polymorphs, taken by the ellipsometry at room temperature. Insets are the corresponding absorption coefficient (α) of the $Ti_2O_3$ polymorphs, collected by the UV–Vis spectroscopy at room temperature. **f** Evolution of the $U$ and $\Delta$ in $Ti_2O_3$ polymorphs. Source data are provided as a Source Data file

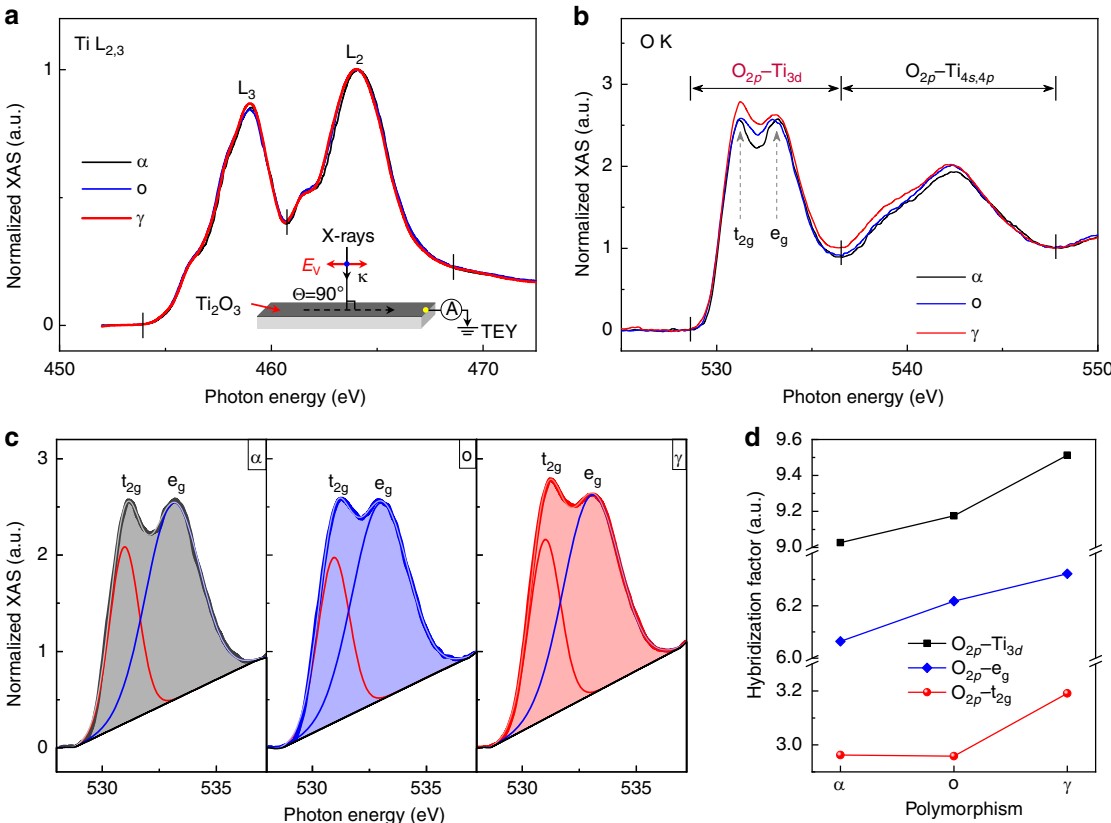

**Fig. 4** Hybridization strength of the Ti–O orbitals in different $Ti_2O_3$ polymorphs. **a** Ti $L_{2,3}$-edge and (**b**) O $K$-edge XAS spectra of the $Ti_2O_3$ polymorphs, collected in the TEY mode at room temperature. The experimental configuration is shown as inset in (**a**). The intensity of the O-$K$ edge XAS spectra in (**b**) are normalized at 547.7 eV. **c** Fitting and (**d**) Integrated intensities of the O $K$-edge pre-edge region (from 528 to 537.2 eV) with subtraction of the linear backgrounds for $Ti_2O_3$ polymorphs. The hybridization factor (H.F.) for $O_{2p}$-$e_g$ and $O_{2p}$-$t_{2g}$ is obtained by calculating the integrated intensities of the fitted $e_g$ (blue lines in **c**) and $t_{2g}$ (red lines in **c**) curves. Source data are provided as a Source Data file

energy positions for each optical transitions, Gaussian-fitting analysis was performed. Similarly, the $a_{1g} - a_{1g}^*$ transitions are still located at around 3 eV, which is consistent with the ellipsometry results. Importantly, the $\Delta$ ($O_{2p} - e_g^\pi$) for α-, o-, and γ-$Ti_2O_3$ polymorphs are verified to be 4.78, 4.48, and 4.37 eV, respectively. As a result, the $\Delta$ decreases in the $Ti_2O_3$ polymorphs with increasing $U$ (Fig. 3f). According to these electronic reconstructions, strong polymorph-dependent properties could be expected in $Ti_2O_3$. It is well known that $\Delta$ is directly related to the hybridization of M–O orbitals, which would further impact on the conductivity[48–50] and electrochemical activity[20,51].

**Hybridization strength of Ti–O orbitals in $Ti_2O_3$ polymorphs**. In order to estimate the hybridization of Ti–O orbitals in $Ti_2O_3$, synchrotron-based X-ray absorption spectroscopy (XAS) was performed to examine the electronic structures of $Ti_2O_3$ polymorphs. The experimental XAS configuration is shown as inset in Fig. 4a. Ti $L_3$ and $L_2$-edge transitions (Fig. 4a) in all $Ti_2O_3$ polymorphs are located at ~458.6 eV and ~463.7 eV, respectively, consistent with the bulk $Ti_2O_3$[56]. The similar Ti $L$-edge XAS spectra obtained from the different $Ti_2O_3$ polymorphs reveal that they are sharing the same $Ti^{3+}$ chemical environment as expected. However, distinct variations are observed at the O $K$-edge XAS spectra. The O $K$-edge XAS spectrum has been widely utilized to study the hybridization of the M–O orbitals in oxides[21,57,58]. Notably, the O $K$-edge pre-edge region near the threshold of the O $K$-edge XAS demonstrates the hybridization between O 2p and transition metal 3d states[21,57]. As shown in

Fig. 4b, two pre-peaks, located at ~531.7 and ~533.7 eV, are used to evaluate the extent of the hybridization of O 2p with Ti 3d $e_g$ and $t_{2g}$ orbitals. Besides the pre-edge, another region of the O $K$-edge at higher energy range corresponds to O 2p orbitals hybridized with Ti 4 s,4p orbitals. Obviously, hybridization of O 2p with Ti 3d and Ti 4 s,4p orbitals are changed in the $Ti_2O_3$ polymorphs duo to their distinct crystal structures.

Integrated intensity of the O $K$-edge pre-edge region with the subtraction of linear backgrounds is commonly used to quantify the hybridization strength of the M–O orbitals in oxides[11,21]. Here, we define a hybridization factor (H.F.) as the integrated intensities of the XAS O $K$-edge pre-edge region (from 528 to 537.2 eV) with the subtraction of the linear backgrounds (shaded area in Fig. 4c). As a consequence, Ti–O hybridization is the strongest in γ-$Ti_2O_3$ with the smallest $\Delta$, while that is the weakest in α-$Ti_2O_3$ with the largest $\Delta$ (Fig. 4d). The same conclusion about the Ti–O hybridization can be obtained by the X-ray photoelectron spectroscopy (XPS) (Supplementary Fig. 8, Supplementary Table 1 and Supplementary Note 4). Stronger Fe 3d –O 2p hybridization was observed in the γ-$Fe_2O_3$ than that in the α-$Fe_2O_3$[59], indicating that the stronger M–O hybridization can be induced by the crystal structure and electronic structure of the γ-phase. As we know, smaller $\Delta$ with stronger hybridization leads to the delocalization of electrons, and thus increases the conductivity of oxides[48–50].

**Electronic transport properties of $Ti_2O_3$ polymorphs**. As expected, various electronic transport behaviors (Fig. 5) are

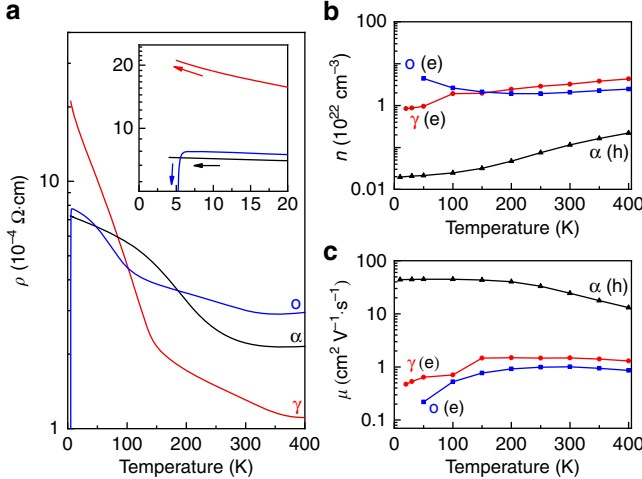

**Fig. 5** Electronic transport properties of $Ti_2O_3$ polymorphs. **a** Resistivity, **b** carrier concentration, and **c** carrier mobility of α-, o-, and γ-$Ti_2O_3$ polymorphs. The inset in (**a**) shows the resistivity at temperatures below 20 K. Source data are provided as a Source Data file

observed in the $Ti_2O_3$ polymorphs with their electronic reconstructions, revealing strong lattice–charge coupling. As shown in Fig. 5a, all $Ti_2O_3$ polymorphs show very low resistivity, and semiconducting behavior is dominant for all below 400 K (except a metal–insulator transition is observed at 360 K for o-$Ti_2O_3$). The kink feature in the resistivity of the newly stabilized γ-$Ti_2O_3$ at around 140 K reveals a semiconductor–semiconductor transition (SST), which is similar to that of o-$Ti_2O_3$[29] and Mg-[$Ti_2$]$O_4$[60] caused by the instability of Ti–Ti dimerization. Interestingly, α-$Ti_2O_3$ shows a nearly flat resistivity below 20 K, while a sudden drop (superconductivity)[29] and continuous increase were observed in that of o-$Ti_2O_3$ and γ-$Ti_2O_3$, respectively (inset of Fig. 5a), indicating the diversity and complexity of $Ti_2O_3$. In addition to the electron–electron interaction, the electron–phonon interaction is also an important factor in determining the electronic behaviors of the $Ti_2O_3$ polymorphs[29,54], which makes $Ti_2O_3$ more complicated and interesting. The change of the resistivity from 400 to 2 K in γ-$Ti_2O_3$ is more than one order of magnitude, which is larger than that of α-$Ti_2O_3$ and o-$Ti_2O_3$. Moreover, the transport behavior of γ-$Ti_2O_3$ is consistent even when its thickness decreases to 20 nm (Supplementary Fig. 9).

Furthermore, the carrier information of the $Ti_2O_3$ polymorphs was investigated by Hall-effect measurement. Interestingly, o-$Ti_2O_3$ and γ-$Ti_2O_3$ are confirmed to be *n*-type semiconducting, whereas α-$Ti_2O_3$ is a *p*-type semiconductor. As shown in Fig. 5b, high electron concentrations at the scale of ~$10^{22}$ cm$^{-3}$ are observed in o-$Ti_2O_3$ and γ-$Ti_2O_3$, which could contribute to the strong Drude absorptions in Fig. 3d, e. Surprisingly, an unexpected increase is observed in the carrier concentration of o-$Ti_2O_3$ at T < 250 K, while that of γ-$Ti_2O_3$ decreases continuously below 400 K. Increase of the carrier concentration in o-$Ti_2O_3$ may be correlated to its superconductivity at lower temperatures, which needs further investigations. Lower carrier concentration (~$10^{21}$ cm$^{-3}$) is observed in α-$Ti_2O_3$, which is about one order of magnitude lower than in o-$Ti_2O_3$ and γ-$Ti_2O_3$. However, the carrier mobility (>10 cm$^2$ V$^{-1}$ s$^{-1}$) in α-$Ti_2O_3$ is about one order of magnitude higher than those in o-$Ti_2O_3$ and γ-$Ti_2O_3$ (Fig. 5c). It should be noted that higher electron concentration leads to more carrier scattering, which results in the lower electron mobility in o-$Ti_2O_3$ and γ-$Ti_2O_3$. Besides, the more significant change of the resistivity (400–2 K) in γ-$Ti_2O_3$

results from the constructive effect between the decreased carrier concentration and mobility, while destructive effects between the carrier concentration and mobility are observed in the α-$Ti_2O_3$ and o-$Ti_2O_3$ that results in the smaller change of the resistivity (400–2 K). More details are shown in Supplementary Note 5. Notably, the carrier concentrations of $Ti_2O_3$ polymorphs at 300 K increases with smaller Δ (Fig. 3f) that result in more delocalized electrons, consistent with the previous scenario[48–50].

**Correlation between HER activity and electronic reconstructions.** In order to elucidate the relationship between the electronic reconstructions and HER catalytic activity, $Ti_2O_3$ polymorphs were used directly as the working electrodes for HER measurements in 0.5 M $H_2SO_4$. The configuration of the HER electrochemical cell is schematically illustrated as the inset of Fig. 6a. Obviously, the electronic reconstructions in $Ti_2O_3$ influenced the HER activities significantly, representing strong polymorph dependence. The influence of the surface microstructure or defect on the observed polymorph-dependent HER performance in $Ti_2O_3$ is discussed and ruled out in Supplementary Note 6. As shown in Fig. 6a, b, the newly stabilized γ-$Ti_2O_3$ has the best activity with the smallest Tafel slope (199 mV dec$^{-1}$), while the bulk-phase α-$Ti_2O_3$ is the least active HER catalyst with the largest Tafel slope (241 mV dec$^{-1}$). Noticeably, the γ-$Ti_2O_3$ displays the smallest overpotential of 271 mV (Fig. 6c) to produce a current density of 10 mA cm$^{-2}$, which is a ~45% reduction compared with that of the α-$Ti_2O_3$ (495 mV). The Tafel plots observed here are a little bit larger than those of the nanostructured oxide samples[22,23], which would be caused by the limited surface area of our film samples. Thus, further improvements could be expected by nanostructure-array fabrications on the films with increased surface areas.

Impressively, the enhanced HER activities are observed in those epitaxially stabilized phases (o- and γ-$Ti_2O_3$) that do not exist in bulk form, which may be applicable to other oxide materials. Hence, efficient TMO-based HER catalysts could be achieved by the selective stabilization of bulk-absent polymorphic phases. Most importantly, the HER activity of the $Ti_2O_3$ polymorphs are strongly correlated to their electronic structures. As shown in Fig. 6d, the overpotentials of HER decreases with smaller Δ accompanied with stronger Ti–O hybridization, unambiguously demonstrating the HER activity is highly enhanced by the electronic reconstructions. Noteworthily, the Faradaic resistances of $Ti_2O_3$ polymorphs, obtained from the electrochemical impendence spectra (Supplementary Fig. 10), do not follow the trend of their HER performances, indicating the enhanced activity is not simply derived from their electrical conductivity, but indeed from the electronic reconstructions. The overall HER catalytic activity of γ-$Ti_2O_3$ may not be comparable with those of the state-of-the-art noble metals (e.g., Pt/C), but considering the tiny weight of the γ-$Ti_2O_3$ film (~100 μg for 300 nm on 1 × 1 cm substrates), the activity of γ-$Ti_2O_3$ is still remarkable as an oxide material. (More discussion is presented in Supplementary Note 7). Moreover, the correlation between the electronic reconstructions and HER activities in the $Ti_2O_3$ polymorphs are robust.

Subsequently, the mechanism behind the electronic-reconstruction enhanced HER activity in $Ti_2O_3$ is explored. In general, a multi-step electrochemical process would take place, during the HER, on the surface of the electrocatalyst, where gaseous hydrogen ($H_2$) is generated. In acid solution, the HER would proceed via either Volmer–Heyrovsky or Volmer–Tafel pathways[61] (Volmer: $H^+ + M + e^- \leftrightharpoons M - H^*$; Heyrovsky: $M - H^* + H^+ + e^- \leftrightharpoons M + H_2$; Tafel: $2\ M - H^* \leftrightharpoons 2M + H_2$, where $H^*$ designates a hydrogen atom chemically adsorbed on the

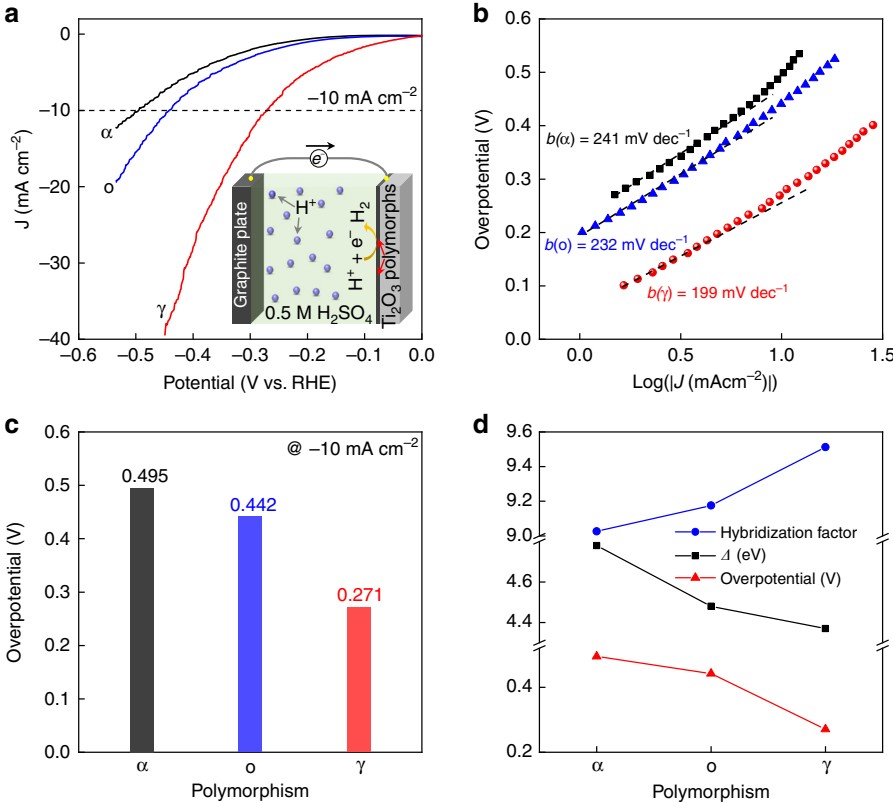

**Fig. 6** Electrocatalytic hydrogen-evolution activities of $Ti_2O_3$ polymorphs. **a** Linear sweep voltammetry (LSV) and (**b**) corresponding Tafel plots of the LSV curves for α-, o-, and γ-$Ti_2O_3$ polymorphs (electrolyte: 0.5 M $H_2SO_4$, scan rate: 5 mV s$^{-1}$). **c** Overpotential of HER for $Ti_2O_3$ polymorphs (@ −10 mA cm$^{-2}$), deduced from (**a**). **d** Correlations between the physical parameters and HER activity in $Ti_2O_3$ polymorphs. Source data are provided as a Source Data file

active sites of the electrocatalyst surface (M)). The hydrogen adsorption free energy ($\Delta G_{H^*}$) is a well-known descriptor for the HER activity, and its optimal value $|\Delta G_{H^*}|$ should be zero, indicating the H$^*$ adsorption is neither too strong nor too weak[61]. Because the H$^*$ adsorption is excessively strong on the oxygen atoms, TMOs are usually considered to be inactive HER electrocatalysts. However, the H$^*$ adsorption on oxides could be weakened by increasing the covalency of the M–O bonds, which results in an activated or enhanced HER efficiency[23].

**DFT simulations**. To gain an in-depth understanding of the observed polymorph-dependent HER in $Ti_2O_3$, we conducted the density functional theory (DFT) simulations to calculate the $\Delta G_{H^*}$ of the Ti sites on the $Ti_2O_3$ polymorphs' surfaces. Since the oxygen atoms cannot provide extra electrons to H, the HER-active sites of $Ti_2O_3$ are the Ti sites ($Ti^{3+}$) with the unpaired $3d^1$ electrons. The optimized models of H adsorbed on the surfaces of $Ti_2O_3$ polymorphs are shown in the Supplementary Fig. 11. H prefers to adsorb at the top site of the Ti atom on the α-$Ti_2O_3$ (0001) surface with a bond length of $d_{H-Ti} = 1.718$ Å, while it prefers to adsorb at the bridge sites of the Ti atoms on the o-$Ti_2O_3$ (011) and γ-$Ti_2O_3$ (001) surfaces resulting in bond lengths of $d_{H-Ti} = 1.908$ Å and $d_{H-Ti} = 1.915$ Å, respectively. The shortest Ti–H bond length on the α-$Ti_2O_3$ (0001) surface implies the strongest bond strength among those adsorption surfaces.

We further analyzed the bond formation between the adsorbed H and Ti atoms by calculating the charge accumulation and depletion around H. The charge density difference is determined by using the formula $\Delta\rho = \rho_{(H+Ti_2O_3)} - (\rho_H + \rho_{Ti_2O_3})$, where $\rho_{(H+Ti_2O_3)}$, $\rho_H$, and $\rho_{Ti_2O_3}$ represent the charge density of the H

adsorbed on the $Ti_2O_3$ surface, isolated H, and the clean $Ti_2O_3$ surface, respectively. The calculated charge density difference for H adsorbed on different $Ti_2O_3$ polymorphs' surfaces are shown in Fig. 7a–c, where the magenta regions show the electron accumulation (bonding states) while the yellow regions show electron depletion (antibonding states). As shown in Fig. 7a–c, there is depletion of electron density on the Ti atoms and accumulation of electron density around the H atoms on all $Ti_2O_3$ surfaces. That is, there is electron charge transfer from Ti to H. The adsorbed H gains electrons from its bonded Ti atoms, leading to the electron accumulation at H and depletion at Ti atoms. Apparently, more yellow regions (electron depletion) accumulate near the adsorbed H on the γ-$Ti_2O_3$ (001) surface (Fig. 7c) than those on the α-$Ti_2O_3$ (0001) and o-$Ti_2O_3$ (011) surfaces, demonstrating the most antibonding states are generated when H adsorbs on the γ-$Ti_2O_3$ (001) surface, which can weaken the adsorption energy of H.

The top valence band of $Ti_2O_3$ (Fig. 3b), just below the Fermi level, is derived from the Ti 3d orbitals ($a_{1g}$ band), whose center is the so called d-band center in the d-band theory[62]. Based on Hammer et al.'s work[62], the energy of the d-band center ($E_d$) with respect to the Fermi level is the critical factor which determines the strength of the interaction between the metal and the adsorbate. Impressively, the $E_d$ of the H-adsorbed α-$Ti_2O_3$ (0001), o-$Ti_2O_3$ (011), and γ-$Ti_2O_3$ (001) surfaces are calculated to be −1.39, −1.53, and −1.97 eV (Supplementary Fig. 12a), respectively. The difference of the d-band center in $Ti_2O_3$ polymorphs can be attributed to their distinct hybridization strength between the Ti 3d and O 2p orbitals. Since the O 2p band is further away from the Fermi level than the Ti 3d ($a_{1g}$) band (Fig. 3b), stronger Ti–O hybridization could make the d-band center downshift

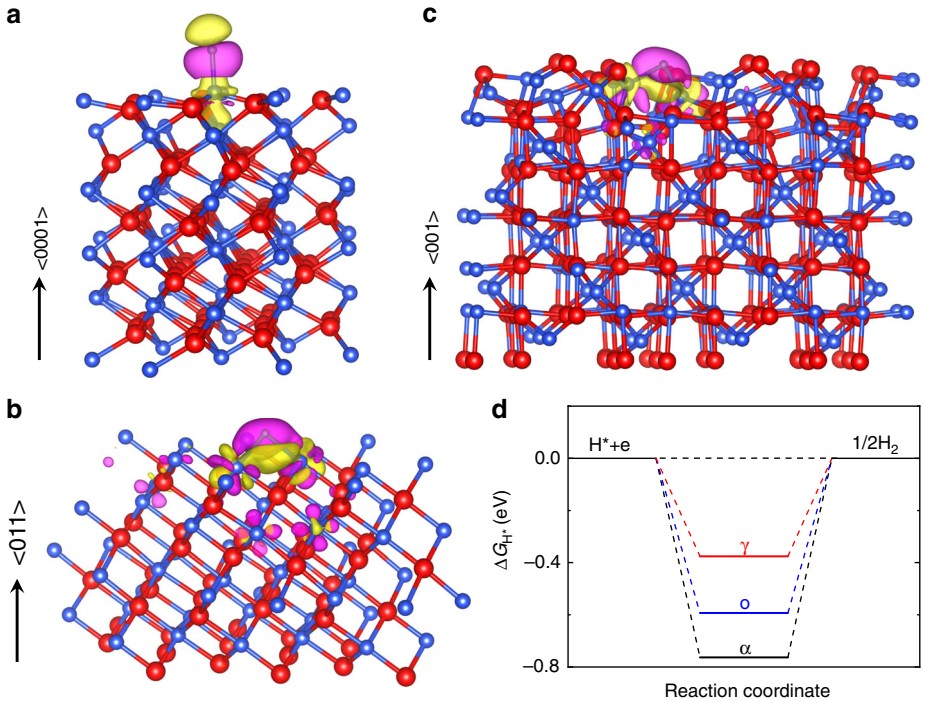

**Fig. 7** DFT simulations. **a–c** Electron charge density differences for H adsorbed on the α-Ti$_2$O$_3$ (0001), o-Ti$_2$O$_3$ (011), and γ-Ti$_2$O$_3$ (001) surfaces, respectively. The surface planes for calculations are chosen based on the experimental orientations of the Ti$_2$O$_3$ epitaxial films (Fig. 1f). The red, blue, and green spheres represent oxygen, titanium, and hydrogen atoms, respectively. The isosurface is taken as 0.002 e Å$^{-3}$. The electron charge accumulation (bonding states) and depletion (antibonding states) are represented by the magenta and yellow regions, respectively. **d** Calculated Gibbs free energy of H on the surfaces of Ti$_2$O$_3$ polymorphs. Source data are provided as a Source Data file

away from the Fermi level. Thus, a lower $d$-band center ($E_d$) is achieved with stronger Ti–O hybridization (Supplementary Fig. 12b). Meanwhile, the lowered $d$-band center will result in an increased filling of the antibonding states[63], which weakens the strength of the Ti–H bond and thus decreases the adsorption energy of H.

Finally, the calculated chemisorption energy of H on the γ-Ti$_2$O$_3$ (001) surface is −0.593 eV, while those on the o-Ti$_2$O$_3$ (011) and α-Ti$_2$O$_3$ (0001) surfaces are −0.809 and −0.979 eV, respectively, which is consistent with the results of the optimized Ti–H bond lengths and the $d$-band center. Considering the correction value of 0.216 eV ($\Delta E_{ZPE} - T\Delta S_H$), $\Delta G_{H^*}$ on the surfaces of Ti$_2$O$_3$ polymorphs are obtained. As shown in Fig. 7d, $\Delta G_{H^*}$ is −0.376, −0.593, and −0.763 eV on the surfaces of γ-Ti$_2$O$_3$ (001), o-Ti$_2$O$_3$ (011), and α-Ti$_2$O$_3$ (0001), respectively, indicating that H has the strongest chemical bonding on the α-Ti$_2$O$_3$ (0001) surface and the weakest chemical bonding on the γ-Ti$_2$O$_3$ (001) surface. Our theoretical simulations are consistent with the experimental results (Fig. 6c), where it can be found that the overpotential is the smallest (0.271 V) on the γ-Ti$_2$O$_3$ (001) surface and the largest (0.495 V) on the α-Ti$_2$O$_3$ (0001) surface. It should be noted that the difference between theoretical and experimental values may arise from the theoretical perfect surface models without considering any intrinsic defects, whereas experimental samples may contain some intrinsic defects on the surfaces. Nevertheless, our calculated $\Delta G_{H^*}$ can act as a descriptor for the experimental HER performance of the Ti$_2$O$_3$ polymorphs as they show the same tendency (Table 1).

## Discussion
In summary, we systematically studied the strong correlations between the polymorphism (lattice symmetry), electronic structure, and HER activity in the epitaxial Ti$_2$O$_3$ films with three

| Table 1 Summary of the parameters of Ti$_2$O$_3$ polymorphs | | | |
|---|---|---|---|
| Polymorphism | α-Ti$_2$O$_3$ | o-Ti$_2$O$_3$ | γ-Ti$_2$O$_3$ |
| Structure | Trigonal | Orthorhombic | Cubic |
| Space group | $R\bar{3}c$ | $Immm$ | $Fd\bar{3}m$ |
| Z | 6 | 2 | 8 |
| Lattice parameters | a = 5.15 Å | a = 9.39 Å | a = 8.53 Å |
| | b = 5.15 Å | b = 4.42 Å | b = 8.53 Å |
| | c = 13.64 Å | c = 2.81 Å | c = 8.53 Å |
| | α = 90° | α = 90° | α = 90° |
| | β = 90° | β = 90° | β = 90° |
| | γ = 120° | γ = 90° | γ = 90° |
| $V$ (Å$^3$) | 313.22 | 116.63 | 620.65 |
| $U$ (eV) | 0.85 | 1.01 | 1.10 |
| $\Delta$ (eV) | 4.78 | 4.48 | 4.37 |
| H.F. | 9.03 | 9.18 | 9.51 |
| S.T. | p | n | n |
| $\rho$ (Ω·cm) | 2.22 × 10$^{-4}$ | 3.01 × 10$^{-4}$ | 1.31 × 10$^{-4}$ |
| $n$ (cm$^{-3}$) | 1.15 × 10$^{21}$ | 2.06 × 10$^{22}$ | 3.23 × 10$^{22}$ |
| $\mu$ (cm$^2$ V$^{-1}$·s$^{-1}$) | 24.44 | 1.01 | 1.48 |
| $E_d$ (eV) | −1.39 | −1.53 | −1.97 |
| $\Delta G_{H^*}$ (eV) | −0.763 | −0.593 | −0.376 |
| Overpotential (V) | 0.495 | 0.442 | 0.271 |

*S.T.* semiconducting type, *H.F.* hybridization factor
The electronic transport parameters are obtained from Fig. 5 at 300 K

different phases. A bulk-absent cubic Ti$_2$O$_3$ polymorph is successfully grown via epitaxial stabilization using PLD, which further extends the polymorphism of Ti$_2$O$_3$. Distinct polymorph-dependent electronic structures and properties are observed in Ti$_2$O$_3$, indicating strong structure–property correlations. More importantly, the electronic reconstructions (varied $U$ and $\Delta$)

observed in the epitaxially stabilized $Ti_2O_3$ polymorphs (orthorhombic and cubic) impact significantly on their HER catalytic activities. Decreased $\Delta$ leads to a substantial enhancement in the HER performance of $\gamma$-$Ti_2O_3$, reducing the overpotential by ~45%, with strongest Ti–O hybridization (lowest $d$-band center). Thus, epitaxial stabilization of bulk-absent polymorphs is demonstrated to be an effective way to discover more efficient HER electrocatalysts in oxides. Moreover, we envision that greater enhancement of the HER activity could be realized by further decreasing the $\Delta$ via doping, or increasing the surface area via nanostructure-array fabrications. Our work provides an unambiguous descriptor for the HER activities of $Ti_2O_3$ polymorphs, which could be applied to other strongly correlated TMO systems.

## Methods

**Fabrication of $Ti_2O_3$ epitaxial films**. All $Ti_2O_3$ films were deposited on single-crystal substrates using pulsed laser deposition (PLD), with a 248 -nm Laser (KrF, Coherent). The thickness of the films is ~300 nm. $\alpha$- and o-$Ti_2O_3$ films were deposited on (0001) $Al_2O_3$ substrates at 600 °C and 900 °C, respectively[29]. The phase separation between $\alpha$- and o-$Ti_2O_3$ was controlled by deposition temperature. $\gamma$-$Ti_2O_3$ films were deposited on (001) $SrTiO_3$ and (001) LAO substrates at 600 °C. The pressure of the PLD chamber was lower than $3.0 \times 10^{-6}$ Torr. The energy density of the laser on the target was fixed to ~2 J cm$^{-2}$. The same corundum $\alpha$-$Ti_2O_3$ target (Sigma-Aldrich 99.99%) was used for all films' deposition.

**Structural characterizations**. X-ray diffraction patterns were recorded using a Bruker D8 DISCOVER high-resolution diffractometer, which is equipped with Cu K$\alpha$ radiation source and LynxEye detector. During the measurements, the diffractometer was operated at 35 kV and 50 mA. In-plane $\varphi$ scans were performed using the synchrotron-based XRD in the Singapore Synchrotron Light Source (SSLS) with a step size of 0.02°. STEM-HAADF images and EELS spectra were collected using the JEOL-ARM200F microscope equipped with an ASCOR aberration corrector, operated at 200 kV. The cross-section TEM samples were prepared by the focused ion beam technique. The EELS line scans were collected with an energy resolution of ~0.1 eV and a spatial resolution of ~0.4 nm.

**Synchrotron-based XAS measurements**. All the XAS data were taken in an ultra-high-vacuum chamber with a base pressure of ~$1 \times 10^{-10}$ mbar at the Surface, Interface and Nanostructure Science (SINS) beam-line[64] of SSLS. The XAS data were obtained by using linear polarized X-rays impinging at an incidence angle of $\Theta = 90°$ from the sample surface (Fig. 4a) at the Ti $L_{2,3}$-edge and O $K$-edge regions. The data were obtained by averaging 20 XAS spectra. All spectra were recorded at room temperature (300 K) using the total electron yield (TEY) mode. The photon energy was calibrated using a standard gold sample in the chamber. In order to record the intrinsic bulk electronic structures of $Ti_2O_3$ polymorphs, samples measured by the synchrotron-based XAS were sputter-cleaned before the measurements.

**Spectroscopic ellipsometry and light absorption measurements**. The ellipsometry parameters $\Psi$ (the ratio between the amplitude of $p$- and $s$-polarized reflected light) and $\Delta$ (the phase difference between of $p$- and $s$-polarized reflected light) were measured using spectroscopic ellipsometer with a photon range of 0.5–4 eV at incident angles 60°, 65°, and 70° at room temperature. The optical conductivity of $Ti_2O_3$ films was extracted from the parameters $\Psi$ and $\Delta$ utilizing an air/$Ti_2O_3$/$Al_2O_3$ (or air/$Ti_2O_3$/$LaAlO_3$) multilayer model, where the $Ti_2O_3$ films were considered as average homogeneous and uniform mediums. The light absorption of the $Ti_2O_3$ films was measured by a UV–Vis spectrophotometer (Shimadzu SolidSpec-3700) in the transmission mode. Backgrounds from the $Al_2O_3$ and $LaAlO_3$ substrates were all subtracted.

**Electronic transport and electrochemical measurements**. The electronic transport properties of the $Ti_2O_3$ films were measured in a Quantum Design physical property measurement system (PPMS). The resistivity and Hall effect were collected using the Van der Pauw geometry. The sample size was $5 \times 5$ mm. Al wires were used to connect the samples and the PPMS puck. The electrochemical measurements were performed in a three-electrode electrochemical cell with a standard VMP3 electrochemical workstation (Bio-logic Inc) at room temperature. During the HER measurements, Hg/HgO and graphite plate were used as the reference and counter electrodes, respectively, while the $Ti_2O_3$ polymorphs films on $10 \times 10$ mm$^2$ single-crystal substrates were acting as the working electrodes directly. The polarization curves were recorded in 0.5 M $H_2SO_4$ with a scan rate of 5 mV s$^{-1}$. All potentials were calibrated with respect to the reversible hydrogen electrode (RHE) using the equation[65]:

$$E_{\text{vsRHE}} = E_{\text{vsHg/HgO}} + 0.059 \times \text{pH} + 0.098 \tag{1}$$

where $E_{\text{vsHg/HgO}}$ was the potential measured against the Hg/HgO reference electrode.

**First-principles calculations**. All the calculations were carried out using DFT + U ($U_{\text{eff}} = U - J = 1.9 - 2.3$ eV)[66] with the generalized Perdew–Burke–Ernzerhof (PBE)[67] and the projector augmented-wave (PAW) pseudopotential plane-wave method[68] as implemented in the VASP code[69]. For the PAW pseudopotentials, we considered $1s^1$ for H, $3d^3 4s^1$ for Ti, and $2s^2 sp^4$ for O. The Monkhorst–Pack (MP) k-point grids of $6 \times 6 \times 2$, $2 \times 6 \times 8$, and $2 \times 2 \times 1$ were used for $\alpha$-$Ti_2O_3$, o-$Ti_2O_3$, and $\gamma$-$Ti_2O_3$ unit cell geometry optimization calculations with a plane-wave basis set with an energy cutoff of 500 eV, respectively. Good convergence was obtained with these parameters, and the total energy was converged to $1.0 \times 10^{-6}$ eV per atom, as well as the stress exerted on the cell is less than 0.1 kbar and the forces exerted on the atoms are less than 0.01 eV/Å. Optimized unit cells were used to build surface models for H Gibbs free energy calculations (Supplementary Fig. 11). For H adsorption calculations, we cleaved a (0001) slab to build the surface model and expand to $2 \times 2 \times 1$ supercell for $\alpha$-$Ti_2O_3$, a (011) slab to build the surface model and expand to $2 \times 2 \times 1$ supercell for o-$Ti_2O_3$, and a (001) slab to build the surface model for $\gamma$-$Ti_2O_3$ according to the experimental results. All the surface models have a vacuum separation of ~15 Å along the z-direction. We carried out calculations with the van der Waals (vdW) correction by employing optPBE-vdW functional[70] using a $2 \times 2 \times 1$ MP k-point grid. The Gibbs free energy of H was calculated by using $\Delta G_{\text{H}^*} = \Delta E_{\text{H}} + \Delta E_{\text{ZPE}} - T\Delta S_{\text{H}}$. Hydrogen chemisorption energy $\Delta E_{\text{H}}$ is computed using $\Delta E_{\text{H}} = E_{\text{(surface+H}^*)} - E_{\text{(surface)}} - \frac{1}{2}E_{\text{H}_2}$, where $E_{\text{(surface+H}^*)}$ and $E_{\text{(surface)}}$ are the total energies of the surface with one adsorbed hydrogen atom and the clean surface, respectively, $E_{\text{H}_2}$ is the energy of hydrogen gas phase. $\Delta E_{\text{ZPE}}$ and $\Delta S_{\text{H}}$ are the differences in zero-point energy (ZPE) and entropy between the adsorbed H$^*$ and gas phase $H_2$. The calculated correction value of $\Delta E_{\text{ZPE}} - T\Delta S_{\text{H}}$ at the temperature ($T$) of 300 K is ~0.216 eV in this study.

## Data availability

All relevant data presented in this paper are available from the authors upon reasonable request. The source data underlying Figs 1f, 3c–f, 4–6, 7d and Supplementary Figs 6, 8-10, 12b are provided as a Source Data file.

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

## Acknowledgements

This work is supported by the Singapore National Research Foundation under CRP Award NRF-CRP10-2012-02, Singapore Ministry of Education under MOE2018-T2-2-043, and AMEIRG18-0022. We acknowledge the Singapore Synchrotron Light Source (SSLS) for providing the facility necessary for conducting the research. The Laboratory is a National Research Infrastructure under the National Research Foundation Singapore. Z.Y. acknowledges supports from the Science and Engineering Research Council of Singapore with Grant no. A1898b0043, and computational resource was provided by A*STAR Computational Resource Centre, Singapore (A*CRC) and the National Supercomputing Centre Singapore (NSCC). X.R.W. acknowledges supports from the Nanyang Assistant Professorship grant from Nanyang Technological University and Academic Research Fund Tier 1 (RG108/17) from Singapore Ministry of Education. We also would like to acknowledge support from the National Research Foundation (NRF) under the Competitive Research Program (NRF-CRP15-2015-01).

## Author contributions

Y.L. and J.S.C. conceived the project. Y.L. designed the experiments, and performed the samples fabrication, characterization, data analysis, and interpretation. L.W. performed the electrochemical measurements. L.M.W., S.W., J.W.C., D.W., and T.V. assisted with the optical measurements. C.S.T., X.M.Y., A.T.S.W., J.W., K.H., and X.R.W. assisted with the XRD measurements. X.J.Y. and M.B.H.B. carried out the XAS measurements. H.W. and S.J.P. assisted with the TEM measurements. Z.G.Y. and Y.W.Z. performed the theoretical calculations. Y.W. and S.D. assisted with the theoretical calculations. Y.L., J.M. X., and J.S.C. discussed the data interpretation. Y.L. and J.S.C. wrote the paper, and all authors commented on the results and the paper.

## Additional information

**Competing interests:** The authors declare no competing interests.

**Peer Review Information:** *Nature Communications* thanks Min-Rui Gao, and the other, anonymous, reviewer(s) for their contribution to the peer review of this work. Peer reviewer reports are available.

