## [Peer Review File · Nature Communications]

Editorial Note: Parts of this peer review file have been redacted as indicated to remove third-party material where no permission to publish could be obtained.

Reviewers' comments:

Reviewer #1 (Remarks to the Author):

Comments:

The present manuscript studies the correlation between the physical electronic structures of Ti₂O₃ polymorphs and their HER activities. Using the epitaxial stabilization methodology, the authors perfectly controlled the generation of three Ti₂O₃ phases and their electronic reconstructions were carefully characterized and analyzed. On the basis of detailed experiments, the authors came to the conclusion that smaller charge-transfer energy with stronger hybridization of Ti 3d and O 2p orbitals weakens the H^{*} adsorption on O atom, giving the improvement of the hydrogen catalytic efficiency (~45% decrease in the overpotential). Although with good presentation, this work has something unclear that hampers its publication at present stage:

1. In general, the carrier mobility of n-type semiconductor is significantly higher than that of p-type one. While in Figure 5, the carrier mobility of α -Ti₂O₃ is much higher than both of γ and σ -Ti₂O₃. What causes this unusual result?

2. Although the authors analyzed the phase structures and electronic reconstructions with details, the surface information of the HER catalysts (such as micro-structure, specific surface area, defects et al.) is largely lacking, which for sure are critical to evaluate an electrocatalyst. Moreover, in another paper of the authors, they described that Ti₂O₃ polymorphs like α and σ -Ti₂O₃ have different types of O vacancies (adfm.201705657). The authors should rule out the possibilities of these defect sites that contribute to the observed HER enhancement.

3. The authors identified that stronger hybridization of Ti 3d and O 2p orbitals weakened the H^{*} adsorption on O atom, enabling an enhancement in the HER efficiency. However, the real experimental or computational evidences regarding this hypothesis are lacking.

4. On page 8&9, the authors studied the hybridization between O 2p and Ti 3d by means of spectroscopic ellipsometry, light absorption and XAS measurements. XPS maybe useful to determine the hybridization states of Ti-O orbitals from the chemical shifts of titanium and oxygen.

5. It is well known that many noble-metal-free HER catalysts have been developed with high activity and stability, some of which even approach the performance of Pt/C benchmark. Why the authors focus their research on Ti₂O₃ materials, which actually are not very competitive. Besides, oxides may readily be reduced in-situ in the potential range of HER, yielding another kind of active material but not Ti₂O₃ itself.

Other minor comments:

6. In line 151 page7, authors used LAO instead of STO to eliminate the effects of the substrate. Could you explain the reasons in detail?

7. In Figure 1c, what does TiO₂-STO(001) view refer to?

Reviewer #2 (Remarks to the Author):

The authors found a method to fabricate a new phase of Ti₂O₃, and investigated it's electrical structure and the hydrogen evolution catalysis. I think this material is very interesting, and the

performance is significant. And it should be admitted that the work on the fabrication of Ti₂O₃ films is rarely. But this work focused on more about the characterization of the material and its performance, and some information about the growth mechanism are lacking. Based on carefully evaluation, I think there are some concerns to be elucidated further before this manuscript to be considered further in Nature Communications.

1. The author claimed that the cubic Ti₂O₃ is new and bulk-absent. Please explain the reason why it's difficult to fabricate this phase in bulk, and why it can be fabricate in film.
2. What does the it means, "By recrystallization on the substrates during epitaxial growth" (line 112, at page 5)? Are there any proof for the structure change of the substrate? Besides, please explain why the change of substrate could lead to different phases of Ti₂O₃.
3. Similar question mentioned above: the author described "The phase separation was controlled by carefully varying deposition temperature and substrates' symmetry", what's the proof for the change of substrates' symmetry in this work?
4. The authors compared the electrical structure of three phases of Ti₂O₃, please collect their differences in a table. It would be more concise and clear.
5. line 237-238, at page 10, the authors described, "The change of the resistivity from 400 to 2 K in γ -Ti₂O₃ is more than one order of magnitude, which is larger than those for α -Ti₂O₃ and o -Ti₂O₃". Please discuss the reason further.
6. This work presented three phases of Ti₂O₃, wherein " α and o " have already fabricated in the other works. Please compare these two phases to those in the other papers.

Reviewers' comments:

Reviewer #1 (Remarks to the Author):

Comments:

The present manuscript studies the correlation between the physical electronic structures of Ti_2O_3 polymorphs and their HER activities. Using the epitaxial stabilization methodology, the authors perfectly controlled the generation of three Ti_2O_3 phases and their electronic reconstructions were carefully characterized and analyzed. On the basis of detailed experiments, the authors came to the conclusion that smaller charge-transfer energy with stronger hybridization of Ti 3d and O 2p orbitals weakens the H^* adsorption on O atom, giving the improvement of the hydrogen catalytic efficiency (~45% decrease in the overpotential). Although with good presentation, this work has something unclear that hampers its publication at present stage:

1. In general, the carrier mobility of n-type semiconductor is significantly higher than that of p-type one. While in Figure 5, the carrier mobility of $\alpha\text{-Ti}_2\text{O}_3$ is much higher than both of γ and o- Ti_2O_3 . What causes this unusual result?

Response: We thank the reviewer for the constructive and helpful comments. We agree with the reviewer that usually the carrier mobility of *n*-type semiconductor is higher than that of the *p*-type ones when their carrier concentrations are similar, e.g. doped silicon (**R1**). It should be noted that the carrier concentration has a significant influence on the carrier mobility. Specifically, the mobility will be lower when the carrier concentration is higher (as shown in **R1**), due to the increased scattering (*Principles of Semiconductor Devices* by Bart Van Zeghbroeck, 2004.). Thus, if the concentration of electrons is much higher than that of the holes, the holes' mobility could be much higher than that of the electrons (e.g. the points marked with circles in **R1**). In our case, the carrier concentrations in γ and o- Ti_2O_3 (Figure 5b) are much higher (~ 1 order high) than that of the $\alpha\text{-Ti}_2\text{O}_3$, indicating much more scattering in γ and o- Ti_2O_3 than that in $\alpha\text{-Ti}_2\text{O}_3$, which would decrease the electron mobility in γ and o- Ti_2O_3 . Therefore, higher hole mobility was achieved in the $\alpha\text{-Ti}_2\text{O}_3$ with a much lower carrier concentration.

[Redacted]

Devices by Bart Van Zeghbroeck, 2004.)

Actually, the carrier concentration of oxides is also related to the metal-oxygen hybridization, while stronger hybridization will delocalize more electrons (**J. Zaanen, *et al. Phys. Rev. Lett.* 55, 418–421 (1985). J. B. Goodenough. *J. Appl. Phys.* 37, 1415 (1966).**), which is consistent with our results. Since our study is focused on different oxide polymorphs, their electron-phonon coupling/scattering must be different due to their different crystal structures, which may also contribute to the carrier mobility of the Ti_2O_3 polymorphs.

Our results are actually usual with the significant carrier concentrations' difference, consistent with the previous observations and theory. We added some clarifications in **page 11, line 23, and page 12, lines 1-2** in the revised manuscript.

2. Although the authors analyzed the phase structures and electronic reconstructions with details, the surface information of the HER catalysts (such as micro-structure, specific surface area, defects *et al.*) is largely lacking, which for sure are critical to evaluate an electrocatalyst. Moreover, in another paper of the authors, they described that Ti_2O_3 polymorphs like α and β - Ti_2O_3 have different types of O vacancies (adfm.201705657). The authors should rule out the possibilities of these defect sites that contribute to the observed HER enhancement.

Response: We thank the reviewer for this important comment. Firstly, it should be noted that all the Ti_2O_3 samples were fabricated by pulsed laser deposition (PLD) technique (**Figure 1a**), resulting in the dense, flat, and fully covered film/substrates heterostructures (**R2**).

R2. Schematic of the film/substrate heterostructure fabricated by PLD.

Atomic force microscopy (AFM) was used to characterize the surface microstructures of the Ti_2O_3 films. As shown in **R3**, the root mean square (RMS) roughness for all Ti_2O_3 samples are quite similar and close to 1 nm, which is very small with respect to their thickness (~ 300 nm), indicating the surfaces of the films are very flat. Thus, there would not be a big difference in the surface area (or specific surface area) of the Ti_2O_3 film samples.

R3. (a-c) AFM images of the α -Ti₂O₃, o-Ti₂O₃, and γ -Ti₂O₃ films, respectively. The image size is 2 × 2 μ m. (Added as Supplementary Figure 2)

As for the defects, in our previous work (Y. Li, *et al. Adv. Funct. Mater.* **28**, 1705657 (2018).), those samples were deposited in very high vacuum ($\sim 3.0 \times 10^{-9}$ Torr) in order to create some oxygen vacancies in the samples. In this present work, we reduced the vacuum to around 3.0×10^{-6} Torr to suppress the formation of oxygen vacancies. Since the Ti₂O₃ films were deposited at high temperatures (above 600 °C), they could be oxidized to TiO₂ with a small amount of O₂ at the O₂ partial pressure of 5.0×10^{-4} Torr (supporting information of Y. Li, *et al. Adv. Funct. Mater.* **28**, 1705657 (2018).). Depending on these experience, 3.0×10^{-6} Torr was confirmed to be an optimized vacuum condition to get pure-phase of Ti₂O₃ films with minimized formation of oxygen vacancies. Since we kept the vacuum at the same value ($\sim 3.0 \times 10^{-6}$ Torr) when we fabricated all the samples in this work, the oxygen vacancies in the Ti₂O₃ polymorphs should be suppressed.

R4. Ti L_{2,3}-edge XAS spectra of the Ti₂O₃ polymorphs, collected in the TEY mode at room temperature (Figure 4a).

Synchrotron-based XAS was used to check the defects in the Ti₂O₃ polymorphs. As shown in **R4** (Figure 4a), the Ti L₃, and L₂-edge XAS spectra of all Ti₂O₃ polymorphs are very similar, and located at ~ 458.6 and ~ 463.7 eV respectively, which are consistent with the bulk Ti₂O₃

samples (K. Han, *et al. Sci. Rep.* **6**, 25455 (2016).), demonstrating the Ti^{3+} chemical environments in the Ti_2O_3 polymorphs.

R5. (a). Ti 2p and (b) O 1s XPS spectra collected from Ti_2O_3 polymorphs. (Added as Supplementary Figure 6)

X-ray photoelectron spectroscopy (XPS) was also performed to characterize the defects on films' surfaces (R5). As expected, the Ti 2p and O 1s spectra of the Ti_2O_3 polymorphs are almost identical (with slight shifts, which will be discussed in the response for comment 4). The positions of the XPS peaks of Ti $2p_{3/2}$ and $2p_{1/2}$, located around 458.4 and 464.0 eV respectively, are consistent with those of the Ti_2O_3 bulk single crystals (R. L. Kurtz, *et al. Surf. Sci. Spectra* **5**, 179 (1998).), indicating the Ti^{3+} chemical environments in those Ti_2O_3 polymorphs. By now, both XAS and XPS results confirm the Ti_2O_3 polymorphs are sharing the same Ti valence state of 3+, with minimized oxygen vacancies.

Thus, with determinations of almost the same roughness (surface area) and Ti valence state in the Ti_2O_3 polymorphs, we can rule out these possible effects which may contribute to the observed polymorph-dependent HER performance. We added the details to the Supplementary Note 3, which is mentioned and highlighted in page 5, lines 18-22. and page 12, lines 15-17 in the revised manuscript.

3. The authors identified that stronger hybridization of Ti 3d and O 2p orbitals weakened the H^* adsorption on O atom, enabling an enhancement in the HER efficiency. However, the real experimental or computational evidences regarding this hypothesis are lacking.

Response: We thank the reviewer for this constructive and valuable comment. To gain an in-depth understanding of the observed polymorph-dependent HER in Ti_2O_3 , we conducted the density functional theory (DFT) simulations to calculate the Gibbs free energy of H, ΔG_{H^*} , of the Ti sites on the Ti_2O_3 polymorphs' surfaces. Since the oxygen atoms cannot provide extra electrons to H, the HER active sites of Ti_2O_3 are Ti sites (Ti^{3+}) with the unpaired $3d^1$ electrons. The

optimized models of H adsorbed on the surfaces of Ti_2O_3 polymorphs are shown in **R6**. H prefers to adsorb at the top site of the Ti atom on the $\alpha\text{-Ti}_2\text{O}_3$ (0001) surface with a bond length of $d_{\text{H-Ti}} = 1.718 \text{ \AA}$, while it prefers to adsorb at the bridge sites of the Ti atoms on the $\text{o-Ti}_2\text{O}_3$ (011) and $\gamma\text{-Ti}_2\text{O}_3$ (001) surfaces resulting in bond lengths of $d_{\text{H-Ti}} = 1.908 \text{ \AA}$ and $d_{\text{H-Ti}} = 1.915 \text{ \AA}$, respectively. The shortest Ti-H bond length on the $\alpha\text{-Ti}_2\text{O}_3$ (0001) surface implies the strongest bond among those adsorption surfaces.

R6. Optimized models of H adsorbed on the surfaces of Ti_2O_3 polymorphs. (a) $\alpha\text{-Ti}_2\text{O}_3$ (0001) surface. (b) $\text{o-Ti}_2\text{O}_3$ (011) surface. (c) $\gamma\text{-Ti}_2\text{O}_3$ (001) surface. The red, blue and green spheres represent oxygen, titanium and hydrogen atoms, respectively. (Added as Supplementary Figure 9)

We further analysed the bond formation between the adsorbed H and Ti atoms by calculating the charge accumulation and depletion around H. The charge density difference is determined by using the formula $\Delta\rho = \rho_{(\text{H}+\text{Ti}_2\text{O}_3)} - (\rho_{\text{H}} + \rho_{\text{Ti}_2\text{O}_3})$, where $\rho_{(\text{H}+\text{Ti}_2\text{O}_3)}$, ρ_{H} and $\rho_{\text{Ti}_2\text{O}_3}$ represent the charge density of the H adsorbed on the Ti_2O_3 surface, isolated H and the clean Ti_2O_3 surface, respectively. The calculated charge density difference for H adsorbed on different Ti_2O_3 polymorphs' surfaces are shown in **R7** a-c, where the magenta regions show the electron accumulation (bonding states) while the yellow regions show electron depletion (anti-bonding states). As shown in **R7** a-c, there is depletion of electron density on the Ti atoms and accumulation of electron density around the H atoms on all Ti_2O_3 polymorphs' surfaces. That is, there is electron charge transfer from Ti to H. The adsorbed H gain electrons from its bonded Ti atoms, leading to the electron accumulation at H and depletion at Ti atoms. Apparently, more yellow regions (electron depletion) accumulate near the adsorbed H on the $\gamma\text{-Ti}_2\text{O}_3$ (001) surface (**R7** c) than those on the $\alpha\text{-Ti}_2\text{O}_3$ (0001) and $\text{o-Ti}_2\text{O}_3$ (011) surfaces, demonstrating the most anti-bonding states are generated when H adsorbed on the $\gamma\text{-Ti}_2\text{O}_3$ (001) surface, which can weaken the adsorption energy of H.

R7. DFT simulations. (a-c) Electron charge density differences for H adsorbed on the α -Ti₂O₃ (0001), o-Ti₂O₃ (011), and γ -Ti₂O₃ (001) surfaces, respectively. The red, blue and green spheres represent oxygen, titanium and hydrogen atoms, respectively. The isosurface is taken as 0.002 e/Å³. The electron charge accumulation (bonding states) and depletion (anti-bonding states) are represented by the magenta and yellow regions, respectively. (d) Calculated Gibbs free energy of H on the surfaces of Ti₂O₃ polymorphs. (Added as Figure 7)

The top valence band of Ti₂O₃ (Figure 3b), just below the Fermi level, is made by the Ti 3*d* orbitals (*a*_{1*g*} band), whose center is the so called *d*-band center in the *d*-band theory (B. Hammer, *et al. Phys. Rev. Lett.* **76**, 2141 (1996)). Based on Hammer, *et al.*'s work, it can be seen that the energy of the *d*-band center (*E*_{*d*}) with respect to the Fermi level is the critical factor which determines the strength of the interaction between the metal and the adsorbate. Impressively, the *E*_{*d*} of the H-adsorbed α -Ti₂O₃ (0001), o-Ti₂O₃ (011) and γ -Ti₂O₃ (001) surfaces are calculated to be -1.39, -1.53 and -1.97 eV (R8 a), respectively. The difference of the *d*-band center in Ti₂O₃ polymorphs could be attributed to their distinct hybridization strength between the Ti 3*d* and O 2*p* orbitals. Since the O 2*p* band is further away from the Fermi level than the Ti 3*d* (*a*_{1*g*}) band (Figure 3b), stronger Ti-O hybridization could make the *d*-band center downshift away from the Fermi level. Thus, lower *d*-band center (*E*_{*d*}) is achieved with stronger Ti-O hybridization (R8 b). Meanwhile, the lowered *d*-band center will result in an increased filling of the antibonding states (Z. Chen, *et al. Angew. Chem. Int. Ed.* **57**, 5076 (2018).), which weakens the strength of the Ti-H bond and thus decreases the adsorption energy of H.

R8. Theoretical calculation of the d -band center for Ti₂O₃ polymorphs. (a) Schematic presentation of the d -band center for different Ti₂O₃ polymorphs. (b) Correlation between the Ti-O hybridization and d -band center. (Added as Supplementary Figure 10)

Finally, the calculated chemisorption energy of H on the γ -Ti₂O₃ (001) surface is -0.593 eV, while those on the o-Ti₂O₃ (011) and α -Ti₂O₃ (0001) surfaces are -0.809 and -0.979 eV respectively, which is consistent with the results of the optimized Ti-H bond lengths and the d -band center. Considering the correction value of 0.216 eV ($\Delta E_{ZPE} - T\Delta S_H$), ΔG_{H^*} on the surfaces of Ti₂O₃ polymorphs are obtained. As shown in R7 d, ΔG_{H^*} is -0.376, -0.593, and -0.763 eV on the surfaces of γ -Ti₂O₃ (001), o-Ti₂O₃ (011) and α -Ti₂O₃ (0001) respectively, indicating that H has the strongest chemical bonding on the α -Ti₂O₃ (0001) surface and the weakest chemical bonding on the γ -Ti₂O₃ (001) surface. Our theoretical simulations are consistent with the experimental results (Figure 6c), where it can be found that the overpotential is the smallest (0.271 V) on the γ -Ti₂O₃ (001) surface and the largest (0.495 V) on the α -Ti₂O₃ (0001) surface. It should be noted that the difference between theoretical and experimental values may arise from the theoretical perfect surface models without considering any intrinsic defects, whereas experimental samples may contain some intrinsic defects on the surfaces. Nevertheless, our calculated ΔG_{H^*} can act as a descriptor for the experimental HER performance of the Ti₂O₃ polymorphs with the same tendency.

The theoretical results and method were added to the revised manuscript, and highlighted in page 14, lines 9-23, page 15, lines 1-22, page 16, lines 1-15, and page 19, lines 5-25.

4. On page 8&9, the authors studied the hybridization between O 2p and Ti 3d by means of spectroscopic ellipsometry, light absorption and XAS measurements. XPS maybe useful to determine the hybridization states of Ti-O orbitals from the chemical shifts of titanium and oxygen.

Response: We thank the reviewer for this helpful comment. Following the suggestion, XPS was performed to characterize the Ti₂O₃ polymorphs. As shown in R5 and T1, compared to that

of α -Ti₂O₃, the Ti 2p_{3/2} and 2p_{1/2} main peaks for α -Ti₂O₃ and γ -Ti₂O₃ are slightly shifted to lower binding energy, while the O 1s peaks are slightly shifted to higher binding energy. Moreover, the binding energy of the Ti 2p_{3/2} main peak in γ -Ti₂O₃ is the lowest, demonstrating the strongest Ti 3d - O 2p hybridization (T. Fujii, *et al. Phys. Rev. B* **59**, 3195 (1999)). Similarly, stronger Fe 3d - O 2p hybridization was observed in the γ -Fe₂O₃ than that of the α -Fe₂O₃ (T. Fujii, *et al. Phys. Rev. B* **59**, 3195 (1999)), indicating that the stronger M-O hybridization can be induced by the crystal structure and electronic structure of the γ -phase.

Polymorphism	Ti 2p _{3/2} (eV)	Ti 2p _{1/2} (eV)	O 1s (eV)
α -Ti ₂ O ₃	458.41	464.06	530.26
α -Ti ₂ O ₃	458.35	463.95	530.30
γ -Ti ₂ O ₃	458.20	463.73	530.48

T1. The peak positions of XPS spectra in **R5**. (Added as Supplementary Table 1)

Furthermore, the XPS results are consistent with our XAS (Figure 4) results, confirming the same trend of the Ti-O hybridization strength in the Ti₂O₃ polymorphs. We added **R5** and **T1** to the revised manuscript, mentioned and highlighted in page 10, lines 12-16.

5. It is well known that many noble-metal-free HER catalysts have been developed with high activity and stability, some of which even approach the performance of Pt/C benchmark. Why the authors focus their research on Ti₂O₃ materials, which actually are not very competitive. Besides, oxides may readily be reduced in-situ in the potential range of HER, yielding another kind of active material but not Ti₂O₃ itself.

Response: We thank the reviewer for this comment. Indeed, several diverse groups of materials including transition metal sulfides (D. Voiry, *et al. Nat. Mater.* **12**, 850 (2013). D. Voiry, *et al. Nano Lett.* **13**, 6222 (2013)), selenides (D. Kong, *et al. J. AM. Chem. Soc.* **136**, 4897 (2014)), and phosphides (H. Yang, *et al. Nano Lett.* **15**, 7616 (2015). P. Xiao, *et al. Adv. Energy Mater.* **5**, 1500985 (2015)) have been exploited as efficient noble-metal-free HER catalysts with different benefits and drawbacks (Y. Yan, *et al. J. Mater. Chem. A* **4**, 17587 (2016). S. Anantharaj, *et al. ACS Catal.* **6**, 8069 (2016). D. Ha, *et al. Nano Energy* **29**, 37-45 (2016)).

Recently, transition metal oxides are attracting more attention as an efficient HER catalyst (X. Xu, *et al. Adv. Mater.* **28**, 6442 (2016). T. Ling, *et al. Nat. Commun.* **8**, 1509 (2017). Y. Zhu, *et al. Nat. Commun.* **10**, 149 (2019)) with compositional flexibility and environmental friendliness, whereas they were usually regarded as the HER inactive materials before, because of the unsuitable adsorption energy of H on the oxygen atoms. By now, the mechanism of the observed HER activity in oxides is still not clear. Thus, new mechanism of HER, based on the

oxides' characteristics, e.g. metal-oxygen hybridization, needs to be explored for the oxide HER electrocatalysts.

Ti₂O₃ shows excellent structural flexibility (Y. Li, *et al. Adv. Funct. Mater.* **28**, 1705657 (2018).) and tunable electronic structures with relatively high and polymorph-dependent HER performances, which provides a wonderful platform to study the HER mechanism of oxides. By investigating the electronic structures and HER in different Ti₂O₃ polymorphs via experimental and theoretical methods, strong correlation between the Ti-O hybridization and HER performance was established in this work. Stronger hybridization of Ti 3*d* and O 2*p* orbitals lowered the *d*-band center of Ti, which weakens the H* adsorption, resulting in the enhanced HER efficiency.

We believe this conclusion could be extended to other transition metal oxide systems. Thus, based on the investigation of Ti₂O₃, a new strategy is provided to further improve the HER performance of oxides by increase the metal-oxygen hybridization, which will help for exploring more efficient oxide HER catalysts. Besides, since the samples in this work are epitaxial films with flat surfaces (**R3**), which is quite different from those of the other groups of materials discussed above, the HER active area of the Ti₂O₃ (flat films) is actually limited. Thus, we believe the HER activity of the Ti₂O₃ (films) could be further enhanced by increasing the surface area via nanostructure-array fabrications. As mentioned by the reviewer, possible reduction of oxides during the HER in acid could be a problem when applying oxide HER catalysts. More research efforts are definitely needed upon these issues in the near future.

We added the details to the Supplementary Note 4. Some clarifications were added to the revised manuscript, and highlighted in page 3, lines 19-21, page 4, lines 6-9, and page 4, lines 13-16.

We thank the reviewer very much for the important and constructive comments, which helped us to revise the manuscript with strong theoretical support.

Other minor comments:

6. In line 151 page 7, authors used LAO instead of STO to eliminate the effects of the substrate. Could you explain the reasons in detail?

Response: We thank the reviewer for this comment. Yes, we used γ -Ti₂O₃/LAO samples instead of γ -Ti₂O₃/STO samples in the electrical and optical measurements, in order to eliminate the effects of STO. The reasons are followed.

Actually, STO and LAO are both widely used substrates for epitaxial growth of oxide films. They have the same perovskite structure and similar lattice constants with wide band gaps ($E_g(\text{STO}) \approx 3.3$ eV (K. van Benthem, *et al. J. Appl. Phys.* **90**, 6156 (2001).), $E_g(\text{LAO}) \approx 6.5$ eV (Y.Y. Mi, *et al. Appl. Phys. Lett.* **90**, 181925 (2007)), making them transparent and insulating. However, for STO, oxygen vacancies could be formed inside of STO when annealing in vacuum at high temperatures (Z. Q. Liu, *et al. Phys. Rev. B* **87**, 220405(R) (2013)). Moreover, the oxygen vacancies in the STO could make the originally insulating STO conductive with induced Ti^{3+} ($3d^1$) (S. A. Lee, *et al. Sci. Rep.* **6**, 23649 (2016)). Meanwhile, the oxygen vacancies could also enhance the light absorption of STO, making the originally transparent STO dark at the visible range (S. A. Lee, *et al. Sci. Rep.* **6**, 23649 (2016)), sharing the similar scenario with TiO_2 (H.H. Pham, *et al. Phys.Chem.Chem.Phys.* **2015**, **17**, 541. S. Chen, *et al. Nanomaterials (Basel)*, **2018**, **8**, 245.). Since the deposition temperature for $\gamma\text{-Ti}_2\text{O}_3$ is 600 °C, oxygen vacancies might be formed in the STO substrates during the film growth. Thus, in order to eliminate the effects of STO in the electrical and optical measurements, we used the $\gamma\text{-Ti}_2\text{O}_3/\text{LAO}$ samples.

For LAO, oxygen vacancies are much more difficult to be formed, which enables LAO keeping insulating and transparent after growing the film. Take advantages of the consistence of LAO, we could subtract the signal from LAO substrates easily when we measure the electrical and optical properties of $\gamma\text{-Ti}_2\text{O}_3/\text{LAO}$ samples. This is why we used LAO instead of STO to eliminate the possible oxygen vacancy effects from the substrates. We added some clarification in the revised manuscript, and highlighted in page 7, lines 9-12.

7. In Figure 1c, what does $\text{TiO}_2\text{-STO}$ (001) view refer to?

Response: We thank the reviewer for pointing this out. As shown in **R9**, along with the $\langle 001 \rangle$ direction, STO is made by alternative SrO layers and TiO_2 layers. Thus, the surface of STO substrates could be the TiO_2 layer or the SrO layer. Since the STO substrates used in our experiments were all etched by hydrofluoric acid and then annealed at 950 °C for 3 hours, the SrO layer would be removed. Thus, the surface of the STO substrate is TiO_2 layer terminated. The process we used to deal with the STO substrates is a well-known and commonly utilized method to obtain the TiO_2 layer terminated STO substrates (S. A. Chambers, *et al, Surf. Sci.* **2012**, **606**, **554**.). Hence, $\text{TiO}_2\text{-STO}$ refers to “ TiO_2 layer terminated STO”, as shown in **R9** (a). $\text{TiO}_2\text{-STO}$ (1) view refers to “view the TiO_2 layer terminated STO from the $\langle 001 \rangle$ direction (Figure 1c)”. **R9** was added as the Supplementary Figure 3, which is mentioned and highlighted in page 25, Figure 1c legend, in the revised manuscript.

R9. (a) and (b) are schematic presentations of TiO₂-terminated and SrO-terminated STO, respectively. (Added as Supplementary Figure 3)

Reviewer #2 (Remarks to the Author):

The authors found a method to fabricate a new phase of Ti₂O₃, and investigated its electrical structure and the hydrogen evolution catalysis. I think this material is very interesting, and the performance is significant. And it should be admitted that the work on the fabrication of Ti₂O₃ films is rarely. But this work focused on more about the characterization of the material and its performance, and some information about the growth mechanism are lacking. Based on carefully evaluation, I think there are some concerns to be elucidated further before this manuscript to be considered further in Nature Communications.

1. The author claimed that the cubic Ti₂O₃ is new and bulk-absent. Please explain the reason why it's difficult to fabricate this phase in bulk, and why it can be fabricated in film.

Response: We thank the reviewer for the constructive and helpful comments, which is very important for the epitaxial growth of oxide films. Firstly, it should be noted that the trigonal α -Ti₂O₃ is the thermodynamic stable phase of Ti₂O₃ and exists in bulk nature, while the cubic γ -Ti₂O₃ is a metastable phase that has not been reported or fabricated in the bulk-form.

High-pressure high-temperature measurements are usually used to study the polymorphism of materials, while the polymorphic transitions could be detected by XRD or Raman (Ono S, et al. *J. Phys. Chem. Solids* **65**, 1527 (2004). Ono S, et al. *J. Phys. Chem. Solids* **66**, 1714 (2005). I. O. Sergey, et al, *J. Phys. Condens. Matter.* **22**, 375402 (2010). S. H. Shim, et al. *Phys. Rev. B* **69**, 144107 (2004).). For Ti₂O₃, as shown in R10 (P-T phase diagram), only one golden orthorhombic phase was fabricated at high temperatures and high pressures (V. O. Sergey, et al,

J. Phys. Condens. Matter. 22, 375402 (2010). V. O. Sergey, et al, Phys. Rev. B 88, 184106 (2013).) Thus, the formation energy of γ -Ti₂O₃ should be higher than that of the golden phase or much more energy is needed for the polymorphic transition. This is why the γ -Ti₂O₃ is difficult to fabricate in the bulk form.

R10: Pressure-temperature phase diagram of Ti₂O₃. (V. O. Sergey, et al, Phys. Rev. B 88, 184106 (2013).)

However, the epitaxial growth of oxides in the film form on the single crystal substrates is a different scenario. Kindly please be reminded that pulsed laser deposition (PLD) technique was used to fabricate the Ti₂O₃ polymorphs in this work. As shown in **R11**, an ultraviolet (UV: 248 nm) pulsed laser beam (pulse duration, 10-50 ns) is focused with a high energy density of $\sim 2 \text{ J/cm}^2$ onto the spinning α -Ti₂O₃ target. The laser pulse is absorbed by the target, and the energy of laser is converted to thermal energy. Rapid heating and vaporization of the target material occurs at the focusing area, generating an expanding plasma plume which contains atoms, molecules, and ions, in both ground and excited states, as well as energetic electrons (**D. H. Lowndes, et al. Science 273, 898 (1996).**). The atoms and ions undergo collisions in the high-density region near the target to create a highly directional expansion perpendicular to the target surface with initial velocities above 10^6 cm/s , then propagates with gradually decreased velocity to the heated substrates (6 cm away from the target). The atoms and ions will nucleate, and then grow into the epitaxial films with different orientations or phases on the surface of the substrates, depending on the nature (orientation, structure, and symmetry) of the substrates and its temperature (**D. H. Lowndes, et al. Science 273, 898 (1996).**). Thus, a **recrystallization** process occurs on the surface of the substrates during growth. Moreover, with tunable kinetic energy of the atoms and ions inside of the plume via controlling the laser energy, the *P-T* phase diagram of the target material is changed during the film deposition process on the substrate's surface, making it possible to fabricate new bulk-absent

metastable phases in the film form with the confinement of substrates (O. Y. Gorbenko, *et al. Chem. Mater.* **14**, 4026-4043 (2002).).

R11: Schematic of the PLD technique (Figure 1a).

During the epitaxial growth, the single crystalline substrates act as the seed crystal, the deposited film will lock into the same in-plane symmetry and crystallographic orientation with respect to the substrate crystal, forming the coherent or semicoherent film/substrate interface (O. Y. Gorbenko, *et al. Chem. Mater.* **14**, 4026-4043 (2002).). Since the energy of the coherent and semicoherent interfaces is significantly lower than that of the noncoherent ones (T2, A. P. Sutton; R. W. Balluffi. *Interfaces in Crystalline Materials*; Calendron Press: Oxford, 1995.), the interfaces formed can affect the choice of the nucleus (film) crystallographic structure because the system tends to minimize the free energy to reach the equilibrium state. Thus, we can control the orientation of the films using different orientated single crystalline substrates (Er-Jia Guo, *et al. ACS Appl. Mater. Interfaces* **9**, 19307–19312 (2017).), and we can also control the crystallographic structure of the film via changing the in-plane symmetry of the substrates' surface (J. H. Lee, *et al. Adv. Mater.* **18**, 3125–3129 (2006).).

[Redacted]

T2: Ranges of solid-solid interface energies for three types of planar interfaces. (A. P. Sutton; R. W. Balluffi. *Interfaces in Crystalline Materials*; Calendron Press: Oxford, 1995.)

Thus, the crystallographic structure realized in the epitaxial films can be different from that of the equilibrium bulk material, which is called “epitaxial stabilization” (O. Y. Gorbenko, *et al. Chem. Mater.* **14**, 4026-4043 (2002). A. Kaul, *et al. J. Cryst. Growth* **275**, e2445 (2005).). From the thermodynamic point of view, epitaxial stabilization demands the change of equilibrium phase due to the epitaxial growth. In particular, the change of the pressure (P)-temperature (T) diagrams is realized for the epitaxial stabilization of metastable polymorphs (O. Y. Gorbenko, *et al. Chem. Mater.* **14**, 4026-4043 (2002).).

R12: a-b In-plane views of the film/substrate interfaces (separately) for α -Ti₂O₃/Al₂O₃, and γ -Ti₂O₃/STO, respectively (Figure 1c and e).

In our work, the trigonal α -Ti₂O₃ was grown on trigonal α -Al₂O₃ single crystalline substrates with the same (trigonal) in-plane symmetry (**R12 a**), forming the coherent film/substrate interface. When we stabilize the cubic γ -Ti₂O₃ films, we change the substrate from trigonal Al₂O₃ to cubic SrTiO₃ (STO). Hence, the cubic in-plane symmetry (**R12 b**) on the STO substrate surfaces ensure that the γ -Ti₂O₃ film grows epitaxially in the cubic phase by maintaining the coherent film/substrate interface and thereby minimizing the interface energy (**T2**). This is why γ -Ti₂O₃ could be fabricated in film form. We added the growth mechanism to the Supplementary Note 1, which is mentioned and highlighted in page 5, lines 17-18.

2. What does it mean, "By recrystallization on the substrates during epitaxial growth" (line 112, at page 5)? Are there any proof for the structure change of the substrate? Besides, please explain why the change of substrate could lead to different phases of Ti₂O₃.

Response: We thank the reviewer for this comment. Kindly please refer to our response to the first comment for more details. As mentioned above, the recrystallization occurs at the surface of the substrates with the atoms and ions inside of the plasma plume. We change the substrate from trigonal Al₂O₃ to cubic STO for stabilization of the γ -Ti₂O₃. Finally, the trigonal α -Ti₂O₃ was grown on trigonal α -Al₂O₃ single crystalline substrates, while cubic γ -Ti₂O₃ was fabricated on cubic STO substrates, forming the coherent film/substrate interfaces.

[Redacted]

R13: a,b Extended in-plane views of the (0001) Al₂O₃ and (011) α -Ti₂O₃. (Y. Li, *et al.* *NPG Asia Mater.* **10**: 522-532 (2018).)

As for the stabilization of the o-Ti₂O₃, even though the in-plane symmetry for (011) o-Ti₂O₃ and (0001) Al₂O₃ is orthorhombic and trigonal from one unit cell view (Figure 1 d), which seems like a noncoherent interface. However, if we extend the unit cells (R13), we can find a semicoherent interface between the (011) o-Ti₂O₃ and (0001) Al₂O₃. It is well known that the trigonal Al₂O₃ also has the hexagonal symmetry (R13 a) if we extend its Z number from 2 to 6. For (011) o-Ti₂O₃ (R13 b), the in-plane symmetry is slightly “distorted hexagonal” from the multi-unit cells view (Y. Li, *et al. NPG Asia Mater.* **10: 522-532 (2018)**). As a result, a “semicoherent” interface was formed between the (011) o-Ti₂O₃ and (0001) Al₂O₃. Thus, the orthorhombic o-Ti₂O₃ was stabilized and fabricated on the Al₂O₃ substrates in the film form.

We need to point out that the deposition temperature for α -Ti₂O₃ and γ -Ti₂O₃ is 600 °C, while that for o-Ti₂O₃ is 900 °C, indicating a little bit higher energy is needed to form this “semicoherent” interface (T2). We added the details to the Supplementary Note 1.

3. Similar question mentioned above: the author described "The phase separation was controlled by carefully varying deposition temperature and substrates' symmetry", what's the proof for the change of substrates' symmetry in this work?

Response: We thank the reviewer for this comment. Kindly please refer to our responses to the first and the second comments for more details. For the stabilization of γ -Ti₂O₃, the in-plane symmetry of the substrate was changed from trigonal to cubic when we changed the substrates from trigonal Al₂O₃ to cubic STO (R12).

4. The authors compared the electrical structure of three phases of Ti₂O₃, please collect their differences in a table. It would be more concise and clear.

Response: We thank the reviewer for this helpful reminder. Following the suggestion, a revised table (T3) was added in the revised manuscript in page 32.

Polymorphism	α -Ti ₂ O ₃	o -Ti ₂ O ₃	γ -Ti ₂ O ₃
Structure	Trigonal	Orthorhombic	Cubic
Space group	$R\bar{3}c$	$Immm$	$Fd\bar{3}m$
Z	6	2	8
Lattice parameters	a = 5.15 Å	a = 9.39 Å	a = 8.53 Å
	b = 5.15 Å	b = 4.42 Å	b = 8.53 Å
	c = 13.64 Å	c = 2.81 Å	c = 8.53 Å
	$\alpha = 90^\circ$	$\alpha = 90^\circ$	$\alpha = 90^\circ$
	$\beta = 90^\circ$	$\beta = 90^\circ$	$\beta = 90^\circ$
	$\gamma = 120^\circ$	$\gamma = 90^\circ$	$\gamma = 90^\circ$
V (Å ³)	313.22	116.63	620.65
U (eV)	0.85	1.01	1.10
Δ (eV)	4.78	4.48	4.37
H.F.	9.03	9.18	9.51
S.T.	p	n	n
ρ ($\Omega \cdot \text{cm}$)	2.22×10^{-4}	3.01×10^{-4}	1.31×10^{-4}
n (/cm ³)	1.15×10^{21}	2.06×10^{22}	3.23×10^{22}
μ (cm ² /(V·s))	24.44	1.01	1.48
E_d (eV)	-1.39	-1.53	-1.97
ΔG_{H^*} (eV)	-0.763	-0.593	-0.376
Overpotential (V)	0.495	0.442	0.271

T3: Summary of the parameters of Ti₂O₃ polymorphs. (Added as new Table 1)

5. line 237-238, at page 10, the authors described, "The change of the resistivity from 400 to 2 K in γ -Ti₂O₃ is more than one order of magnitude, which is larger than those for α -Ti₂O₃ and o -Ti₂O₃". Please discuss the reason further.

Response: We thank the reviewer for pointing this out. Since the resistivity of the semiconductor can be shown as $\rho = \frac{1}{q n \mu}$; where q is the elementary charge (1.602×10^{-19} coulombs), n and μ are concentration and mobility of the majority carriers, respectively. Thus, the resistivity is inversely proportional to the carrier concentration and mobility. That is, the resistivity will increase with the carrier concentration and mobility decrease.

Specifically, in α -Ti₂O₃ (Figure 5), the carrier concentration is decreased with decreasing temperature, which will increase the resistivity. However, the mobility is increased at lower temperatures, which will decrease the resistivity. That is, the changes of the carrier concentration and mobility with the reduced temperature have an opposite impact on its resistivity. Thus, the resistivity of α -Ti₂O₃ is increased at low temperatures as a result of the destructive effect between the carrier concentration and mobility. Similarly, in o -Ti₂O₃ (Figure 5), the carrier mobility is

decreased with decreasing temperature, while the carrier concentration is anomaly increased at lower temperatures ($T < 250$ K). Hence, the increase of the resistivity of α - Ti_2O_3 is also a result of the destructive effect between the carrier concentration and mobility.

However, in γ - Ti_2O_3 (Figure 5), both the carrier concentration and mobility are decreased with the temperature decrease, and they have the same impact on the resistivity, that is, they will both make the resistivity increase. So, the increase of the resistivity of γ - Ti_2O_3 is a result of the constructive effect between the carrier concentration and mobility, which results in the larger change of resistivity in γ - Ti_2O_3 .

As for why the carrier concentration and mobility are changed in the different ways in different Ti_2O_3 polymorphs, the electron-electron interactions and electron-phonon interactions will play the important roles in controlling the carriers' behaviors, which is out of the scope of this work and needs further study. We added the details to the **Supplementary Note 2**, which is mentioned and highlighted in **page 12, lines 2-6** in the revised manuscript.

6. This work presented three phases of Ti_2O_3 , wherein " α and α' " have already been fabricated in the other works. Please compare these two phases to those in the other papers.

Response: We thank the reviewer for this comment. Yes, " α and α' " phases of Ti_2O_3 have been fabricated and reported in our previous works (Y. Li, *et al. Adv. Funct. Mater.* **28**, 1705657 (2018). Y. Li, *et al. NPG Asia Mater.* **10**, 522 (2018).). " α and α' " share the same structure with those in our previous studies, but fabricated with different thickness, vacuum and PLD chambers, demonstrating the excellent reproducibility and consistency of Ti_2O_3 films. Kindly please refer to our response to the second comment of reviewer 1 for more details.

The thoughtful and constructive comments from the reviewer are highly appreciated, which guided us to revise the manuscript with more information on the growth mechanism of the Ti_2O_3 polymorphs.

Reviewers' comments:

Reviewer #1 (Remarks to the Author):

The authors have carried out new experiments and added theoretical calculations to address the reviewers' comments. The AFM images of Ti₂O₃ electrocatalysts show no significant difference on surface micro-structure. Moreover, synchrotron XAS and high-resolution XPS demonstrate the similar Ti³⁺ chemical environments in the Ti₂O₃ polymorphs. They also conducted the DFT simulations to calculate the Gibbs free energy of H on the Ti sites on the Ti₂O₃ polymorphs' surfaces. The above results on structural characterizations and calculations match with the observed HER activities: that is, more efficient TMO-based HER electrocatalysts can be gained by enhancing the metal-oxygen hybridization via selective stabilization of polymorph phases.

Based on above, I would suggest that the revised manuscript can be accepted for publication once the following minor questions are addressed:

1. In page 10, the authors defined the integration of the O K-edge pre-edge. They need to specify how the hybridization factor (H.F.) is defined and calculated.
2. According to AFM images, the Ti₂O₃ polymorphs consist of many randomly oriented nanoparticles. In DFT simulations, why the authors chose (001), (011) and (001) planes as the α -Ti₂O₃, α -Ti₂O₃, γ -Ti₂O₃ electrocatalysts' surface for calculation?

Reviewer #2 (Remarks to the Author):

The authors presents detailed response to the first round of reviewing. A few questions still exist and need further explanations.

1. A key issue in this manuscript is the fabrication of cubic Ti₂O₃ film. And the authors insist that this phase is bulk-absent and can be obtained due to the coherent or semicoherent film/substrate interface during epitaxial growth using PLD technique. However, it's strange that the pressure-temperature phase diagram of Ti₂O₃ (Fig. R10) doesn't exhibit the emergence of cubic Ti₂O₃ even at high pressure of GPa scale. If so, why the epitaxial growth could lead to the formation of cubic Ti₂O₃? Although the introduction on the mechanism of PLD growth is abundant, it's difficult to find the answer. By contrast, some other oxides, such as VO₂, the metastable phases could be observed in the pressure-temperature phase diagram.
2. Many other titanium oxides have been developed recently, such as black TiO₂, which has also been proposed to be quite suitable for photocatalysis and Hydrogen Production (Chen et al., *Science*, 2011, 331, 746; Chen et al., *Chem. Soc. Rev.*, 2015, 44, 1861.), and Ti₃O₅ (Ohkoshi, et al., *Nat. Chem.*, 2010, 2, 539; Shen, et al., *Appl. Phys. Lett.*, 2017, 111, 191902), which exhibits polymorphs. The introduction in this manuscript lacks some descriptions on these titanium oxides.

Reviewers' comments:

Reviewer #1 (Remarks to the Author):

Comments:

The authors have carried out new experiments and added theoretical calculations to address the reviewers' comments. The AFM images of Ti₂O₃ electrocatalysts show no significant difference on surface micro-structure. Moreover, synchrotron XAS and high-resolution XPS demonstrate the similar Ti³⁺ chemical environments in the Ti₂O₃ polymorphs. They also conducted the DFT simulations to calculate the Gibbs free energy of H on the Ti sites on the Ti₂O₃ polymorphs' surfaces. The above results on structural characterizations and calculations match with the observed HER activities: that is, more efficient TMO-based HER electrocatalysts can be gained by enhancing the metal-oxygen hybridization via selective stabilization of polymorph phases. Based on above, I would suggest that the revised manuscript can be accepted for publication once the following minor questions are addressed:

1. In page 10, the authors defined the integration of the O K-edge pre-edge. They need to specify how the hybridization factor (H.F.) is defined and calculated.

Response: We thank the reviewer for the positive evaluation and helpful comments.

In this work, the hybridization factor (H.F.) is defined as the integrated intensities of the XAS O-K pre-edge region with subtraction of the linear backgrounds (shaded area in Figure 4c), which is commonly used to quantify the hybridization strength of the metal-oxygen orbitals in oxides (J. Suntivich, *et al. J. Phys. Chem. C* **118**, 1856 (2014). A. Grimaud, *et al. Nat. Commun.* **4**, 2439 (2013)). For the calculation, we first normalized the intensity of the O-K edge XAS spectra at 547.7 eV (as shown in Figure 4b). Then, the H.F. is obtained by calculating the integrated intensities of the O-K pre-edge XAS spectra from 528 to 537.2 eV with the subtraction of the intensity of the linear backgrounds, as shown in Figure 4d (O_{2p}-Ti_{3d}). The H.F. for O_{2p}-e_g and O_{2p}-t_{2g} is obtained by calculating the integrated intensities of the fitted e_g (blue lines in Figure 4c) and t_{2g} (red lines in Figure 4c) curves (528-537.2 eV). We added the details to the revised manuscript and highlighted in page 10, lines 11-13 and page 28 (legend of Figure 4).

2. According to AFM images, the Ti₂O₃ polymorphs consist of many randomly oriented nanoparticles. In DFT simulations, why the authors chose (001), (011) and (001) planes as the α -Ti₂O₃, β -Ti₂O₃, γ -Ti₂O₃ electrocatalysts' surface for calculation?

Response: We thank the reviewer for this comment. In DFT simulations, we choose the planes for calculations based on the experimental results of the orientations of the Ti₂O₃ epitaxial films. As shown in Figure 1f, the orientation of the Ti₂O₃ epitaxial films was confirmed to be <0001>, <011>, and <001> for α -Ti₂O₃, β -Ti₂O₃ and γ -Ti₂O₃, respectively. Although the surfaces of the films are not perfect flat as the theoretical surface models we used, the roughness (~1 nm) is quite small compared to their thickness (~300 nm), revealing the flat surfaces. Yes, the AFM

images show some “island-like” features at the surfaces of the films, which seems like the “nanoparticles” but actually not. It is a common topography for the epitaxial films’ surfaces (S. M. Park, *et al. Nat. Nanotech.* **13**, 366 (2018). V. Ukleev, *et al. Sci. Rep.* **8**, 8741 (2018). W. J. Kim, *et al. Curr. Appl. Phys.* **19**, 400 (2019).) especially when the thickness is above 50 nm.

Since the AFM is a very high-resolution technique to characterize the topography of the surfaces, “island-like” features with even small roughness (<1 nm) can be observed in the AFM images. Actually, they are not randomly nanoparticles. To verify this point, we used the scanning electron microscopy (SEM) and scanning transmission electron microscopy (STEM) to check the surfaces and cross-sections of the films. As shown in **R1**, no nanoparticles were observed in the SEM (with the similar scale of the AFM images) and STEM images. Meanwhile, the SEM and STEM images confirmed the dense Ti₂O₃ films with flat surfaces again. Thus, we believe the planes on the surfaces of the Ti₂O₃ films should be identical to their orientations, that is (0001), (011), and (001) for α -Ti₂O₃, o-Ti₂O₃ and γ -Ti₂O₃, respectively.

R1: (a-c) SEM images collected from the surfaces of α -Ti₂O₃, o-Ti₂O₃, and γ -Ti₂O₃ films, respectively. (d-f) Low-resolution STEM images collected from the cross-sections of α -Ti₂O₃, o-Ti₂O₃, and γ -Ti₂O₃ films, respectively. (Added as Supplementary Figure 3)

Based on the above results, we chose the (0001), (011) and (001) planes as the α -Ti₂O₃, o-Ti₂O₃, γ -Ti₂O₃ electrocatalysts’ surfaces for the theoretical calculations. Some clarifications were added to the revised manuscript in page 6, lines 1-2 and page 31 (legend of Figure 7).

We thank the reviewer very much for the constructive comments, which helped us to revise and improve the manuscript further.

Reviewer #2 (Remarks to the Author):

The authors presents detailed response to the first round of reviewing. A few questions still exist and need further explanations.

1. A key issue in this manuscript is the fabrication of cubic Ti_2O_3 film. And the authors insist that this phase is bulk-absent and can be obtained due to the coherent or semicoherent film/substrate interface during epitaxial growth using PLD technique. However, it's strange that the pressure-temperature phase diagram of Ti_2O_3 (Fig. R10) doesn't exhibit the emergence of cubic Ti_2O_3 even at high pressure of GPa scale. If so, why the epitaxial growth could lead to the formation of cubic Ti_2O_3 ? Although the introduction on the mechanism of PLD growth is abundant, it's difficult to find the answer. By contrast, some other oxides, such as VO_2 , the metastable phases could be observed in the pressure-temperature phase diagram.

Response: We thank the reviewer for the important and helpful comments.

We need to point out that most of the metastable phases epitaxially stabilized in the film-form could be found in the pressure-temperature phase diagram (e.g. VO_2), which is the usual cases. However, there are few unusual cases that is some bulk-absent phases can only be epitaxially stabilized on the substrates in the film-form, such as body-centered cubic (bcc) Co (G. A. Prinz, *Phys. Rev. Lett.* **54**, 1051 (1985).), hexagonal TbMnO_3 (J. H. Lee, *et al. Adv. Mater.* **18**, 3125 (2006).), and orthorhombic Ti_2O_3 (*Immm*) (Y. Li, *et al. Adv. Funct. Mater.* **28**, 1705657 (2018).). This is because the pressure-temperature induced polymorphic transition is a thermodynamic equilibrium process (**R2**) between different structures, while the epitaxial growth (in this study) is a non-equilibrium process with the confinement of the substrates (**R3**). The epitaxial stabilization of the metastable phases in the film-form is a more complicated process than the bulk phase transitions. However, it is a very fascinating method to study the emergent properties of the epitaxially stabilized metastable phases. In order to provide a clear presentation of the epitaxial- growth process, the epitaxial stabilization of the cubic $\gamma\text{-Ti}_2\text{O}_3$ is illustrated in **R3**.

[Redacted]

R2: Schematic presentation of the temperature-pressure induced bulk polymorphic transition from $\alpha\text{-Ti}_2\text{O}_3$ to Th_2S_3 -type Ti_2O_3 (S. V. Ovsyannikov, *et al. Phys. Rev. B* **88**, 184106 (2013).).

As shown in **R3a**, a plume will be generated when the pulsed laser is focused on the target

(α -Ti₂O₃). Meanwhile, the crystal structure of the α -Ti₂O₃ is broken, forming the titanium and oxygen atoms or ions (**R3b**) inside of the plasma plume. When the atoms or ions arrived at the surface of the heated substrates (STO), the in-plane symmetry of the substrate will provide a confinement effect to drive the atoms or ions to form the (cubic γ -Ti₂O₃/cubic STO) coherent interface (**R3c**) on the substrate (cubic STO), minimizing the solid-solid interface energy (σ^{ss}). Based on this cubic/cubic coherent interface, the cubic γ -Ti₂O₃ is epitaxially stabilized on the cubic STO substrate (**R3d**).

R3: Schematic of the epitaxial stabilization of γ -Ti₂O₃. (a) PLD process. (b) Plume formation by pulsed laser ablation. (c) Cubic/cubic coherent interface with low σ^{ss} formed between γ -Ti₂O₃ and STO due to the confinement of the substrate. (d) Epitaxial stabilization of the γ -Ti₂O₃ based on the cubic/cubic coherent interface. (Added as Supplementary Figure 12)

Using the complementary suite of the microscopic (STEM, EELS) and spectroscopic (XRD, XAS, XPS) techniques, the cubic γ -Ti₂O₃ was confirmed to be fabricated in the film-form in this study. It should be mentioned that, in contrast to TiO₂, Ti₂O₃ has received much less attention before. The comprehensive phase diagram of the Ti₂O₃ should be studied further, and more metastable phases might be unearthed in the future, which is out of the scope of this study whereas our future research plans. More explanations were added to the Supplementary Note 1 in the revised manuscript (in page 13 of the supplementary information).

2. Many other titanium oxides have been developed recently, such as black TiO₂, which has also been proposed to be quite suitable for photocatalysis and Hydrogen Production (Chen et al., Science, 2011, 331, 746; Chen et al., Chem. Soc. Rev., 2015, 44, 1861.), and Ti₃O₅ (Ohkoshi, et al., Nat. Chem., 2010, 2, 539; Shen, et al., Appl. Phys. Lett., 2017, 111, 191902), which exhibits polymorphs. The introduction in this manuscript lacks some descriptions on these titanium oxides.

Response: We thank the reviewer for this important reminder. Indeed, TiO₂ has been extensively explored for the photocatalytic water splitting and reduction of CO₂. Plenty of works were reported to enhance its efficiency by increasing its light absorption via doping or reduction (Y. Li, et al. Chem. Mater. 30, 4383 (2018)). Among them, black anatase TiO₂ nanocrystals

exhibited dramatically higher photocatalytic activity of the hydrogen generation than that of the ordinary white TiO₂ (Chen *et al.*, *Science*, **331**, 746 (2011); Chen *et al.*, *Chem. Soc. Rev.* **44**, 1861 (2015)). Moreover, the Ti₃O₅ also show the interesting polymorph-dependent metal-semiconductor transition (Ohkoshi, *et al.*, *Nat. Chem.* **2**, 539 (2010); Shen, *et al.*, *Appl. Phys. Lett.* **111**, 191902 (2017)). Following the suggestions, we added some additional descriptions on the black TiO₂ and the polymorph-dependent properties of the Ti₃O₅ to the introduction part. The works as mentioned above were cited in the revised manuscript (refs. 9,10, 38, 39), and highlighted in page 3, lines 9-12 and page 5, lines 10-12.

We thank the reviewers very much for the helpful comments and reminder, which helped us to revise and further improve the readability of the manuscript.

REVIEWERS' COMMENTS:

Reviewer #1 (Remarks to the Author):

I appreciate and satisfy with the responses from the authors. The revised manuscript now is suitable for being published in Nature Communications.

Reviewer #2 (Remarks to the Author):

The authors presented convincing response to the reviewers' comments. So I recommend this manuscript to be accepted for publication.

REVIEWERS' COMMENTS:

Reviewer #1 (Remarks to the Author):

I appreciate and satisfy with the responses from the authors. The revised manuscript now is suitable for being published in Nature Communications.

Response: We thank the reviewer for the positive support to publication.

Reviewer #2 (Remarks to the Author):

The authors presented convincing response to the reviewers' comments. So I recommend this manuscript to be accepted for publication.

Response: We thank the reviewer for the positive support to publication.